# CFASL: Composite Factor-Aligned Symmetry Learning for Disentanglement in Variational AutoEncoder

**Hee-Jun Jung** *jungheejun93@gm.gist.ac.kr*
*AI Graduate School*
*Gwangju Institute of Science and Technology (GIST)*

**Jeahyoung Jeong** *jeahyoung98@gm.gist.ac.kr*
*AI Graduate School*
*Gwangju Institute of Science and Technology (GIST)*

**Kangil Kim**[*] *kikim01@gist.ac.kr*
*AI Graduate School*
*Gwangju Institute of Science and Technology (GIST)*

**Reviewed on OpenReview:** *https://openreview.net/forum?id=mDGvrH7lju*

## Abstract

Symmetries of input and latent vectors have provided valuable insights for disentanglement learning in VAEs. However, only a few works were proposed as an unsupervised method, and even these works require known factor information in the training data. We propose a novel method, Composite Factor-Aligned Symmetry Learning (CFASL), which is integrated into VAEs for learning symmetry-based disentanglement in unsupervised learning without any knowledge of the dataset factor information. CFASL incorporates three novel features for learning symmetry-based disentanglement: 1) Injecting inductive bias to align latent vector dimensions to factor-aligned symmetries within an explicit learnable symmetry codebook 2) Learning a composite symmetry to express unknown factors change between two random samples by learning factor-aligned symmetries within the codebook 3) Inducing a group equivariant encoder and decoder in training VAEs with the two conditions. In addition, we propose an extended evaluation metric for multi-factor changes in comparison to disentanglement evaluation in VAEs. In quantitative and in-depth qualitative analysis, CFASL demonstrates a significant improvement of disentanglement in single-factor change, and multi-factor change conditions compared to state-of-the-art methods.

## 1 Introduction

Disentangling representations by intrinsic factors of datasets is a crucial issue in machine learning literature (Bengio et al., 2013). In the Variational Autoencoder (VAE) frameworks, a prevalent method to handle the issue is to factorize latent vector dimensions to encapsulate specific factor information (Kingma & Welling, 2013; Higgins et al., 2017; Chen et al., 2018; Kim & Mnih, 2018; Jeong & Song, 2019; Shao et al., 2020; 2022). Although their effective disentanglement learning methods, Locatello et al. (2019) raises the serious difficulty of disentanglement without sufficient inductive bias.

In VAE literature, recent works using group theory offer a possible solution to inject such inductive bias by decomposing group symmetries (Higgins et al., 2018) in the latent vector space. To implement group equivariant VAE, Winter et al. (2022); Nasiri & Bepler (2022) achieve the translation and rotation equivariant VAE. The other branch implements the group equivariant function (Yang et al., 2022; Keller & Welling, 2021b) over the pre-defined group elements. All of the methods effectively improve disentanglement by

---

[*]Corresponding author

adjusting symmetries, but they focused on learning symmetries among observations to inject inductive bias rather than factorizing group elements to align them on a single factor and a single dimension changes, as introduced in the definition provided in Higgins et al. (2018).

In recent works, unsupervised learning approaches of group equivariant models has been introduced. Miyato et al. (2022); Quessard et al. (2020) represent the symmetries on the latent vector space, which correspond to the symmetries on the input space, by considering the sequential observations. Also, Winter et al. (2022) proposes the group invariant and equivariant representations with different modules to learn the different groups of dataset structure. However, these approaches, despite being unsupervised learning, require the factor information of the dataset to construct the sequential input and to set different modules for learning symmetries.

This paper introduces a novel disentanglement method for Composite Factor-Aligned Symmetry Learning (CFASL) within VAE frameworks, aimed at addressing the challenges encountered in unsupervised learning scenarios, particularly the absence of explicit knowledge about the factor structure in datasets. Our methodology follows as 1) a network architecture to learn an explicit codebook of symmetries, responsible for each single factor change, called *factor-aligned* symmetries, 2) training losses to inject inductive bias into an explicit codebook where each factor-aligned symmetry only impacts a single dimension value of the latent vectors for disentangled representations, 3) learning composite symmetries by predicting single factor changes themselves without the information of factor labels for unsupervised learning, 4) implementing group equivariant encoder and decoder functions such that factor-aligned symmetry affects the latent vector space, 5) an extended metric (m-FVM$_k$) to evaluate disentanglement in the multi-factor change condition. We conduct quantitative and qualitative analyses of our method on common benchmarks of disentanglement in VAEs.

Table 1: Notation Table

| Group | | Codebook | |
|---|---|---|---|
| $G$ | Group | $\mathcal{G}$ | Codebook |
| $GL(n, \mathbb{R})$ | General Lie group | $\mathcal{G}^i$ | $i^{th}$ section of codebook |
| $g$ | Group element of Group $G$ | $\mathfrak{g}_j^i$ | $j^{th}$ element of $\mathcal{G}^i$ |
| $\mathfrak{gl}(n, \mathbb{R})$ | Lie algebra of $GL(n, \mathbb{R})$ | $|S|$ | size of section |
| $\mathfrak{g}$ | Lie algebra $\in \mathbb{R}^{n \times n}$ | $|SS|$ | size of subsection |
| $\alpha(\cdot, \cdot)$ | group action | $g_j^i$ | $j^{th}$ symmetry of $i^{th}$ section of codebook, equal to $exp(\mathfrak{g}_j^i)$ |
| | | $g_c$ | composite symmetry |
| **Set** | | | |
| $X$ | dataset | | |
| $\hat{X}$ | subset of dataset | **Others** | |
| $F$ | Set of factors $\{\boldsymbol{f}_1, \ldots \boldsymbol{f}_k\}$ | $\mathbb{R}$ | Real number |
| $f^i$ | $i^{th}$ index value of $f$ | $\sigma$ | standard deviation (scalar) |
| $f^{-i}$ | values of $f$ except $f^i$ | $\boldsymbol{\sigma}$ | standard deviation (vector) |
| $Z$ | Set of latent vectors | $\boldsymbol{\mu}$ | mean (vector) |
| $\boldsymbol{z}$ | latent vector | $\mathbb{X}_{|B|}$ | Random samples with $|B|$ batch size |
| | | $\langle \cdot, \cdot \rangle$ | dot product |
| **Function** | | $| \cdot |_2$ | L2 norm |
| $Gen(\cdot)$ | $F \to X$ | $\perp$ | orthogonal |
| $exp$ | matrix exponential | $\|$ | parallel |

## 2 Preliminaries

### 2.1 Group Theory

**Group:** A group is a set $G$ together with binary operation $\circ$, that combines any two elements $g_a$ and $g_b$ in $G$, such that the following properties:

- closure: $g_a, g_b \in G \Rightarrow g_a \circ g_b \in G$.

- Associativity: $\forall g_a, g_b, g_c \in G,\ s.t.\ (g_a \circ g_b) \circ g_c = g_a \circ (g_b \circ g_c)$.

- Identity element: There exists an element $e \in G$, *s.t.* $\forall g \in G, e \circ g = g \circ e = g$.

- Inverse element: $\forall g \in G, \exists g^{-1} \in G$: $g \circ g^{-1} = g^{-1} \circ g = e$.

**Group action:** Let $(G, \circ)$ be a group and set $X$, binary operation $\cdot : G \times X \to X$, then group action $\alpha : \alpha(g, x) = g \cdot x$ following properties:

- Identity: $e \cdot x = x$, where $e \in G, x \in X$.

- Compatibility: $\forall g_a, g_b \in G, x \in X, (g_a \circ g_b) \cdot x = g_a \cdot (g_b \cdot x)$.

**Equivariant map:** Let $G$ be a group and both $X_1$ and $X_2$ are $G$-set with corresponding group action of $G$ in each sets, where $g \in G$. Then a function $f : X_1 \to X_2$ is equivariant if $f(g \cdot X_1) = g \cdot f(X_1)$.

**Lie Group and Lie algebra:** Lie group is defined as a group that simultaneously functions as a differentiable manifold, with the operations of group multiplication and inversion being smooth and differentiable. In this paper, we consider the matrix Lie group, which is a Lie group realized as a subgroup contained in $GL(n, \mathbb{R})$, the group of $n \times n$ invertible matrices over $\mathbb{R}$.

Lie algebra $\mathfrak{gl}(n, \mathbb{R})$ is a tangent space of the Lie group at the identity element. The Lie algebra covers a Lie group by the matrix exponential of elements of $\mathfrak{gl}(n, \mathbb{R})$ as $exp : \mathfrak{gl}(n, \mathbb{R}) \to GL(n, \mathbb{R})$, where $exp(\boldsymbol{X}) = \boldsymbol{I} + \boldsymbol{X} + \frac{1}{2!}\boldsymbol{X}^2 + \frac{1}{3!}\boldsymbol{X}^3 + \cdots$.

## 2.2 Disentanglement Learning

**Variational Auto-Encoder(VAE)** We take VAE (Kingma & Welling, 2013) as a base framework. VAE optimizes the tractable evidence lower bound (ELBO) instead of performing intractable maximum likelihood estimation as:

$$\text{ELBO} = \mathbb{E}_{q_\phi(z|x)} \log p_\theta(x|z) - D_{\text{KL}}(q_\phi(z|x)||p(z)), \tag{1}$$

where $q_\phi(z|x)$ is the approximated posterior distribution, $q_\phi(z|x) \sim \mathcal{N}(\mu_\phi, \sigma_\phi I)$, the prior $p(z) \sim \mathcal{N}(0, I)$, the first term of Equation 1 is a reconstruction error between input and outputs from the decoder $p_\theta(x|z)$, and second term of Equation 1 is a Kullback-Leibler term to reduce the approximated posterior and prior distance.

**FactorVAE Metric (FVM)** FVM metric (Kim & Mnih, 2018) is one of the disentanglement learning metrics, which evaluates the consistency of dimension variation as a single factor is fixed. First, the fixed factor $f^i$ is selected, and other factors $f^{-i}$ are randomly selected, then generate the subset of the dataset $\hat{X}$, which corresponds to factor $f$, where $\hat{X} = \{x_1, x_2, \ldots, x_i\}$, $x_j = Gen(f_j^i, f_j^{-i})$, and $Gen(\cdot)$ is a generator. The second, latent vector from the encoder of VAE is normalized by the standard deviation ($\sigma$) of the dataset $(\boldsymbol{z}_i \ / \ \sigma)$, then selects the $d^{th}$ dimension as $d = \arg\min_{d'} Var(\boldsymbol{z}_j^{d'})$, $Var(\cdot)$ is a variance of each dimension. The FMV metric is the accuracy of the majority vote classifier with data points $(d, i)$.

## 2.3 Symmetries for Inductive Bias

**Disentangled Representation Based on the Group** Higgins et al. (2018) shows the definition of disentangled representation through the group. First, the generator $Gen : F \to X$ and the inference process $h : X \to Z$ are defined. Then a group $G$ of symmetries acts on $F$ via a group action $\cdot : G \times F \to F$. Lastly, symmetries decompose as a direct product $G = G_1 \times \ldots \times G_n$. Then the disentangled representation is defined with 1) group action $\cdot : G \times Z \to Z$, 2) composition $b : F \to Z$ is an equivariant map, and 3) $Z_i$ is fixed by the action of all $G_j$, and affected only by $G_i$, where decomposition $Z = Z_1 \times \ldots \times Z_n$.

**Implimentation of Equivariant Model** To implement an equivariant map for learning symmetries, previous works utilize the Variational Auto-Encoder with an additional objective function defined as $q_\phi(g^x \cdot_x x) = g^z \cdot_z q_\phi(x)$, where $q_\phi : x \to z$ is an encoder, $g^x, g^z$ is a symmetry on input space $\mathcal{X}$ and latent vector space $\mathcal{Z}$, respectively. Yang et al. (2022); Winter et al. (2022), represent the $g^z$ in the latent vector space

instead $g^x$ with following equation: $q_\phi(x_2) = g^z \cdot_z q_\phi(x_1)$, where $x_2 = g^x \cdot_x x_1$ to represent the $g^z$ correspond to $g^x$. Symmetry $g^z$ is defined as a specific group such as $SO(3)$, $SE(n)$ or $GL(n)$ (Yang et al., 2022; Winter et al., 2022; Zhu et al., 2021).

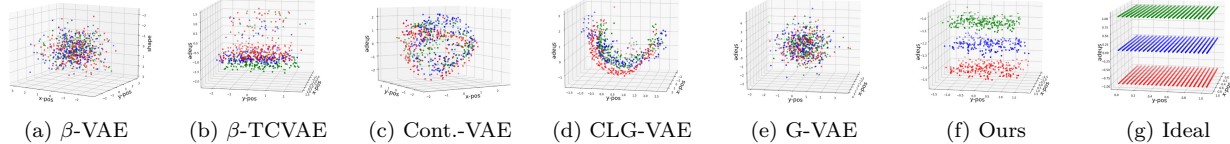

(a) $\beta$-VAE    (b) $\beta$-TCVAE    (c) Cont.-VAE    (d) CLG-VAE    (e) G-VAE    (f) Ours    (g) Ideal

Figure 1: Distribution of latent vectors for dimensions responsible for Shape, X-pos, and Y-pos factors in the dSprites dataset. The groupified-VAE method is applied to $\beta$-TCVAE because this model shows a better evaluation score. The results show disentanglement for shape from the combination of the other two factors by coloring three shapes (square, ellipse, and heart) as red, blue, and green color, respectively. Each 3D plot shows the whole distribution. We fix Scale and Orientation factor values, and plot randomly sampled 640 inputs (20.8% of all possible observations ($32 \times 32 \times 3 = 3,072$)). We select the dimensions responsible for the factors by selecting the largest value of the Kullback-Leibler divergence between the prior and the posterior. Cont.-VAE is a Control-VAE.

## 3 Related Work

**Disentanglement Learning** Diverse works for unsupervised disentanglement learning have elaborated in the machine learning field. The VAE based approaches have factorized latent vector dimensions with weighted hyper-parameters or controllable weighted values to penalize Kullback-Leibler divergence (KL divergence) (Higgins et al., 2017; Shao et al., 2020; 2022). Extended works penalize total correlation for factorizing latent vector dimensions with divided KL divergence (Chen et al., 2018) and discriminator (Kim & Mnih, 2018). Differently, we induce disentanglement learning with group equivariant VAE for inductive bias.

**Group Theory-Based Approaches for Disentangled Representation** In recent periods, various unsupervised disentanglement learning research proposes different approaches with another definition of disentanglement, which is based on the group theory (Higgins et al., 2018). To learn the equivariant function, Topographic VAE (Keller & Welling, 2021a) proposes the sequentially permuted activations on the latent vector space called shifting temporal coherence, and Groupified VAE (Yang et al., 2022) method proposes that inputs pass the encoder and decoder two times to implement permutation group equivariant VAE models. Also, Commutative Lie Group VAE (CLG-VAE) (Zhu et al., 2021; Mercatali et al., 2022) maps latent vectors into Lie algebra with one-parameter subgroup decomposition for inductive bias to learn the group structure from abstract canonical point to inputs. Differently, we propose the trainable symmetries that are extracted between two samples directly on the latent space while maintaining the equivariance function between input and latent vector space.

**Symmetry Learning with Equivariant Model** Lie group equivariant CNN (Dehmamy et al., 2021; Finzi et al., 2020) construct the Lie algebra convolutional network to discover the symmetries automatically. In the other literature, several works extract symmetries, which consist of matrices, between two inputs or objects. Miyato et al. (2022) extracts the symmetries between sequential or sequentially augmented inputs by penalizing the transformation of difference of the same time interval. Other work extracts the symmetries by comparing two inputs, in which the differentiated factor is a rotation or translation, and implements symmetries with block diagonal matrices (Bouchacourt et al., 2021). Furthermore, Marchetti et al. (2023) decomposes the class and pose factor simultaneously by invariant and equivariant loss function with weakly supervised learning. The unsupervised learning work (Winter et al., 2022) achieves class invariant and group equivariant function in less constraint conditions. Differently, we generally extend the a class invariant and group equivariant model in the more complex disparity condition without any knowledge of the factors of datasets.

| method | Inductive Bias | | |
|---|---|---|---|
| | dataset info. | learnable symmetry | orthogonality |
| (Higgins et al., 2017; Chen et al., 2018; Kim & Mnih, 2018) | ✗ | ✗ | ✗ |
| (Zhu et al., 2021; Miyato et al., 2022; Winter et al., 2022) | ✓ | ✓ | ✗ |
| Ours | ✗ | ✓ | ✓ |

Table 2: Summary of injected inductive bias of models for disentanglement learning. *dataset info.* represents that methods need the information of factors of the dataset. *learnable symmetry* represents the learnable parameters to represent the symmetries. *orthogonality* represents the orthogonality between the axes and latent vectors through the group action.

# 4 Limits of Disentanglement Learning of VAE

By the definition of disentangled representation (Bengio et al., 2013; Higgins et al., 2018), the disentangled representations are distributed on the flattened surface as shown in Fig. 1g because each change of the factor only affects a single dimension of latent vector. However, the previous methods (Higgins et al., 2017; Chen et al., 2018; Shao et al., 2020; Zhu et al., 2021; Yang et al., 2022) show the entangled representations on their latent vector space as shown in Fig. 1a-1c. Even though the group theory-based methods improve the disentanglement performance (Zhu et al., 2021; Yang et al., 2022), these still struggle with the same problem as shown in Fig. 1d and 1e. In addition, symmetries are represented on the latent vector space for disentangled representations. In current works (Miyato et al., 2022; Keller & Welling, 2021b; Quessard et al., 2020), the sequential observation is considered in unsupervised learning. However, these works need the knowledge of sequential changes of images to set up inputs manually, as summarized in Table 2.

To enhance these two problems of disentanglement learning of group theory-based methods, addressing two questions is crucial:

1. Do the explicitly defined symmetries impact the structuring of a disentangled space as depicted in Fig. 1g?

2. Can these symmetries be represented through unsupervised learning without any prior knowledge of factor information?

# 5 Methods

Our work is mainly focused on how to 1) define the inputs and symmetry, 2) optimize the symmetry, 3) represent the composite symmetry, and 4) inject the symmetry as an inductive bias. We first define the 1) inputs as a pair of two samples, 2) group, group action, and $G$-set, and 3) codebook in section 5.1. In the next, we optimize the codebook for disentangled representation in section 5.2, then extract the composite symmetry to represent transformation between two inputs in section 5.3. Lastly, we introduce the objective loss to inject an inductive bias for disentangled representations in section 5.4.

## 5.1 Inputs and Symmetry

**Input: A Pair of Two Samples** To learn the symmetries between inputs with unknown factors changes, we randomly pair the two samples as an input. During the training, samples in the mini-batch $\mathbb{X}_{|B|}$ are divided into two parts $\mathbb{X}^1_{|B|} = \{x^1_1, x^1_2, \ldots, x^1_{\frac{|B|}{2}}\}$, and $\mathbb{X}^2_{|B|} = \{x^2_1, x^2_2, \ldots, x^2_{\frac{|B|}{2}}\}$, where $|B|$ is a mini-batch size. In the next, our model pairs the samples $(x^1_1, x^2_1), (x^1_2, x^2_2), \ldots, (x^1_{\frac{|B|}{2}}, x^2_{\frac{|B|}{2}})$ and is used for learning symmetries between the elements of each pair.

**Group, Group action and $G$-set** We define the symmetry group as a matrix general Lie group $GL(n)$, and set group action $\alpha$ as $\alpha : \alpha(g, \boldsymbol{z}) = \boldsymbol{z}^\intercal g$, where $g \in GL(n)$, $\boldsymbol{z} \in \mathbb{R}^n$, and latent vector space $\mathcal{Z}$ is a $G$-set.

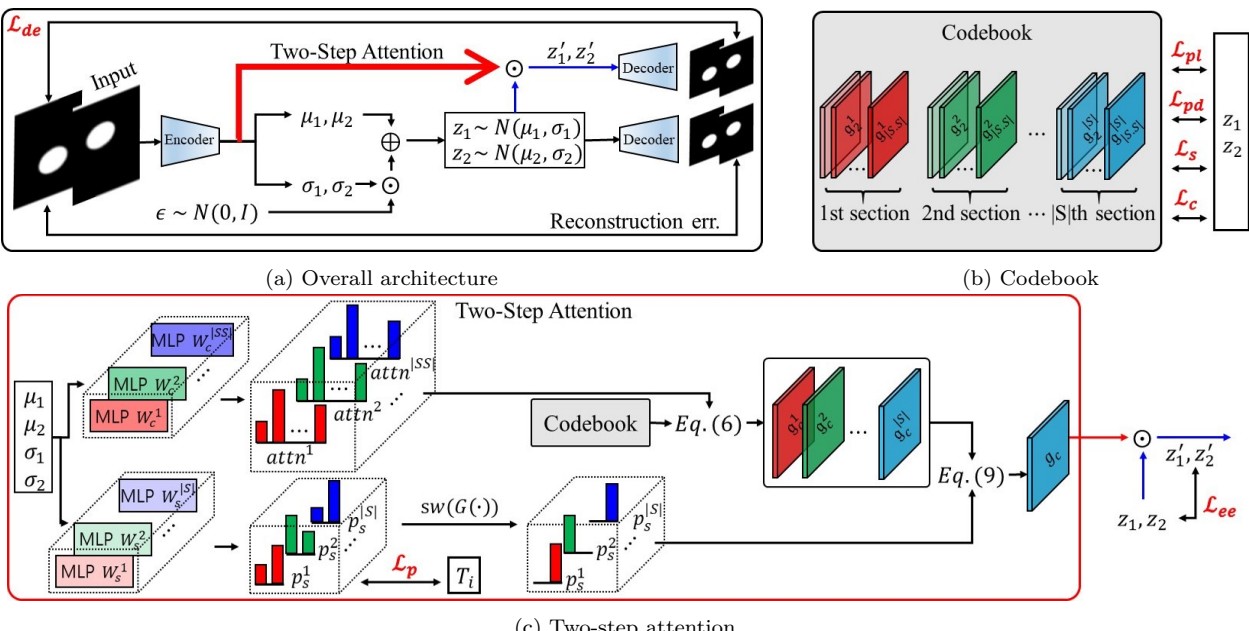

(a) Overall architecture        (b) Codebook

(c) Two-step attention

Figure 2: The overall architecture of the proposed method. $\longleftrightarrow$ refers to a loss function. 1) A pair of images (*e.g.*, *differences* between two images are in the x- and y-position) is given, and the goal of the model is to represent the *differences* on the latent vector space, called the *composite symmetry* $g_c$ for disentangled representations. 2) The *codebook* is designed to represent the composite symmetry $g_c$. 3) Each section of the codebook is separated to affect a single factor *e.g.*, the $i^{th}$ section affects the x-position, and the $j^{th}$ section affects the y-position of images. 4) Each section consists of Lie algebra to provide diversity of symmetries. 5) As shown in (b), each loss optimizes the codebook to guarantee the 3) as follows: i) symmetries from the same section affect the same factor through the parallel loss $\mathcal{L}_{pl}$ (*e.g.*, symmetries from $i^{th}$ section $exp(\mathfrak{g}_k^i)$ only affects the x-position), ii) each section affects different factors by the perpendicular loss $\mathcal{L}_{pd}$ (*e.g.*, symmetry from $i^{th}$ and $j^{th}$ section $g_c^i$ and $g_c^j$ affect x-position and y-position respectively), and iii) each section changes a single dimension of latent vectors for disentangled representation by the sparsity loss $\mathcal{L}_s$. 6) *attn* ensures diversity in symmetries representation, and $p_s$ predicts the activated section, in this case, $i^{th}$ and $j^{th}$ sections for x- and y-position differences. 7) The model then represents the composite symmetry $g_c$. 8) Lastly, model optimizes the $\mathcal{L}_{ee}$ to match $g_c z_1(= z_2')$ and $z_2$, and $\mathcal{L}_{de}$ to match the $x_1$ and $p_\theta(g_c \circ q_\phi(x_2))$ to inject the inductive bias.

**Codebook: Explicit and Learnable Symmetry Representation for** $GL(n)$    To allow the direct injection of inductive bias into symmetries, we implement an explicit and trainable codebook for symmetry representation. we consider the symmetry group on the latent vector space as a subgroup of the general lie group $GL(n)$ under a matrix multiplication. The codebook $\mathcal{G} = \{\mathcal{G}^1, \mathcal{G}^2, \ldots, \mathcal{G}^k\}$ is composed of sections $\mathcal{G}^i$, which affect to a different single factor, where $k \in \{1, 2, \ldots |S|\}$, and $|S|$ is the number of sections. The section $\mathcal{G}^i$ is composed of Lie algebra $\{\mathfrak{g}_1^i, \mathfrak{g}_2^i, \ldots, \mathfrak{g}_l^i\}$, where $\mathfrak{g}_j^i \in \mathbb{R}^{|D| \times |D|}$, $l \in \{1, 2, \ldots, |SS|\}$, $|SS|$ is the number of elements in each section, and $|D|$ is a dimension size of latent $\boldsymbol{z}$. We assume that each Lie algebra consists of linearly independent bases $\mathfrak{B} = \{\mathfrak{B}_i | \mathfrak{B}_i \in \mathbb{R}^{n \times n}, \sum_i \alpha_i \mathfrak{B}_i \neq 0, \alpha_i \neq 0\}$: $\mathfrak{g}_j^i = \sum_b \alpha_b^{i,j} \mathfrak{B}_b$, where $b \in \{1, 2, \ldots, kl\}$. Then, the dimension of the element of the codebook is equal to $|\mathfrak{B}|$, and the dimension of the Lie group composed of the codebook element is also $|\mathfrak{B}|$. To utilize previously studied effective expression of symmetry for disentanglement, we set the symmetry to be continuous (Higgins et al., 2022) and invertible via matrix exponential form (Xiao & Liu, 2020) as $g_j^i = \mathbf{e}^{\mathfrak{g}_j^i} = \sum_{k=0}^{\infty} \frac{1}{k!}(\mathfrak{g}_j^i)^k$ to construct the Lie group (Hall, 2015).

## 5.2 Codebook: Factor-Aligned Symmetry Learning with Inductive Bias (Orthogonality)

We define the *factor-aligned symmetry* that represents a corresponding factor change on the latent vector space and each independent factor only affects a single dimension value of the latent vector. For factor-aligned symmetry, we compose the symmetry codebook and inject inductive bias via 1) parallel loss $\mathcal{L}_{pl}$ and

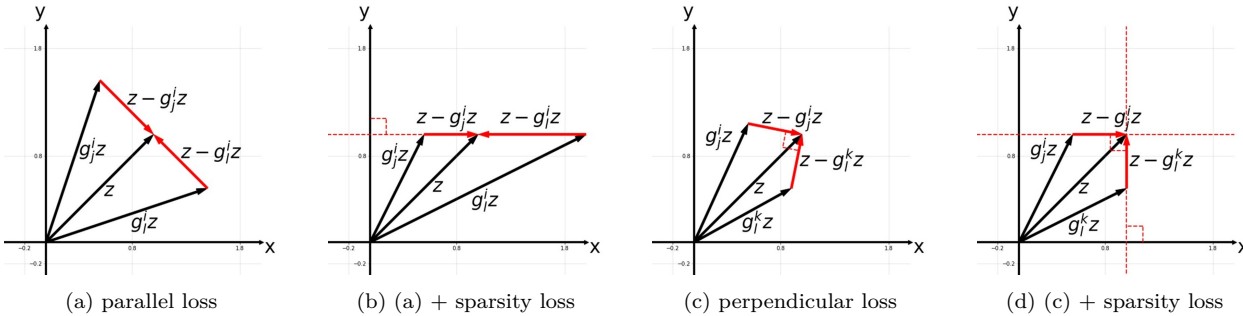

(a) parallel loss  (b) (a) + sparsity loss  (c) perpendicular loss  (d) (c) + sparsity loss

Figure 3: Roles of parallel, perpendicular, and sparsity loss on symmetries in the codebook for adjusting representation change. Parallel loss is for symmetries of the same section, and perpendicular loss is for different sections. Each axis (x and y) only affects a single factor.

2) perpendicular loss $\mathcal{L}_{pd}$ that match each symmetry to a single factor changes. Each loss optimizes the symmetries to affect the same and different factors, respectively. Then we add 3) sparsity loss $\mathcal{L}_s$ to the losses for disentangled representations as shown in Fig. 3. It aligns a single factor change to an axis of latent vector space. Also, we implement the 4) commutative loss $\mathcal{L}_c$ to reduce the computational costs for matrix exponential multiplication.

**Inductive Bias: Group Elements of the Same Section Impacts on the Same Factor Changes**
As we design the symmetries from the same section of the codebook to affect the same factor shown in Fig. 2 (example), we add a bias that the latent vector is changed by symmetries of the same section to be parallel ($\boldsymbol{z} - g_j^i \boldsymbol{z} \parallel \boldsymbol{z} - g_l^i \boldsymbol{z}$ for $i$th section) as shown in Fig. 3a. We define a loss function to make them parallel as:

$$\mathcal{L}_{pl} = \sum_{i=1}^{|S|} \sum_{j,k=1}^{|SS|} \log \frac{\langle \boldsymbol{z} - g_j^i \boldsymbol{z}, \boldsymbol{z} - g_k^i \boldsymbol{z} \rangle}{|\boldsymbol{z} - g_j^i \boldsymbol{z}|_2 \cdot |\boldsymbol{z} - g_k^i \boldsymbol{z}|_2}, \tag{2}$$

where $g_j^i = \mathbf{e}^{\mathfrak{g}_j^i}$, $\langle \cdot, \cdot \rangle$ is a dot product, and $| \cdot |_2$ is a L2 norm.

**Inductive Bias: Group Elements of the Different Section Impacts on the Different Factor Changes**  As shown in Fig. 2 example of perpendicular loss to enforce each section affects different factor, we inject another bias that changes the latent vector through symmetries of the each sections to be orthogonal for different factors change impacts ( $\boldsymbol{z} - g_j^i \boldsymbol{z} \perp \boldsymbol{z} - g_l^k \boldsymbol{z}$ for different $i$th and $k$th sections) as shown in Fig. 3c. The loss for inducing the orthogonality is

$$\mathcal{L}_{pd} = \sum_{i,k=1, i \neq k}^{|S|} \sum_{j,l=1}^{|SS|} \frac{\langle \boldsymbol{z} - g_j^i \boldsymbol{z}, \boldsymbol{z} - g_l^k \boldsymbol{z} \rangle}{|\boldsymbol{z} - g_j^i \boldsymbol{z}|_2 \cdot |\boldsymbol{z} - g_l^k \boldsymbol{z}|_2}. \tag{3}$$

Due to the expensive computational cost for Eq. 3 ($O(|S|^2 \cdot |SS|^2)$), we randomly select a $(j, l)$ pair of symmetries of each section. This random selection still holds the orthogonality, because if all elements in the same section satisfy Equation 2 and a pair of elements from a different section $(\mathcal{G}^i, \mathcal{G}^j)$ satisfies Equation 3, then any pair of the element ($\mathfrak{g}^i \in \mathcal{G}^i$, $\mathfrak{g}^j \in \mathcal{G}^j$) satisfies the Equation 3. More details are in the Appendix B.

**Inductive Bias: Align Each Factor Changes to the Axis of Latent Space for Disentangled Representations**  Factorizing latent dimensions to represent the change of independent factors is an attribute of disentanglement defined in Bengio et al. (2013) and derived by ELBO term in VAE training frameworks (Chen et al., 2018; Kim & Mnih, 2018). However, the parallel and the perpendicular loss do not factorize latent dimension as shown in Fig. 3a, 3c. To guide symmetries to hold the attribute, we enforce the change $\Delta_j^i \boldsymbol{z} = \boldsymbol{z} - g_j^i \boldsymbol{z}$ to be a parallel shift to a unit vector as Fig. 3b, 3d via sparsity loss defined as

$$\mathcal{L}_s = \sum_{i=1}^{|S|} \sum_j^{|SS|} \left[ \Big[ \sum_{k=1}^{|D|} (\Delta_j^i \boldsymbol{z}_k)^2 \Big]^2 - \max_k ((\Delta_j^i \boldsymbol{z}_k)^2)^2 \right], \tag{4}$$

where $\Delta_j^i \boldsymbol{z}_k$ is a $k^{th}$ dimension value.

**Commutativity Loss for Computational Efficiency** In the computation of the composite symmetry $g_c$, the production $\prod_{i=1}^{|S|} \hat{g}_c^i$ is computationally expensive because the Taylor series repeated for all $(i, j)$ pairs. To reduce the cost by repetition, we enforce all pairs of basis $\mathfrak{g}_i^j$ to be commutative to convert the production to $\mathbf{e}^{\sum_i \mathfrak{g}_c^i}$ (By the matrix exponential property: $\mathbf{e}^{\boldsymbol{A}} \mathbf{e}^{\boldsymbol{B}} = \mathbf{e}^{\boldsymbol{A} + \boldsymbol{B}}$ as $\boldsymbol{A}\boldsymbol{B} = \boldsymbol{B}\boldsymbol{A}$, where $\boldsymbol{A}, \boldsymbol{B} \in \mathbb{R}^{n \times n}$). The loss for the commutativity is defined as:

$$\mathcal{L}_c = \sum_{i,k=1}^{|S|} \sum_{j,l=1}^{|SS|} \mathfrak{g}_j^i \mathfrak{g}_l^k - \mathfrak{g}_l^k \mathfrak{g}_j^i \to \mathbf{0}. \tag{5}$$

### 5.3 Composition of Factor-Aligned Symmetries via Two-Step Attention for Unsupervised Learning

In this section, we introduce how the model extracts the symmetries between two inputs. We propose the a two-steps process as follows: 1) the model extracts the Lie algebra of each section $\mathfrak{g}_c^i$ to represent the factor-aligned symmetries between two inputs in the *first step*, then 2) predicts which section of the codebook is activated in the *second step* process as shown in Fig. 2 example.

**First Step: Select Factor-Aligned Symmetry** In the first step, the model generates the factor-aligned symmetries of each section through the attention score, as shown in Fig. 2c:

$$\mathfrak{g}_c^i = \sum_{j=1}^{|SS|} attn_j^i \mathfrak{g}_j^i, \tag{6}$$

where $attn_j^i = softmax([M; \Sigma]\boldsymbol{W}_c^i + \boldsymbol{b}_c^i)$ $[M; \Sigma] = [\boldsymbol{\mu}_1; \boldsymbol{\sigma}_1; \boldsymbol{\mu}_2; \boldsymbol{\sigma}_2]$, $\boldsymbol{W}_c^i \in \mathbb{R}^{4|D| \times |SS|}$ and $\boldsymbol{b}_c^i \in \mathbb{R}^{|SS|}$ are learnable parameters, $i \in \{1, 2, \dots |S|\}$, and $softmax(\cdot)$ is a softmax function.

**Second Step: Section Selection** In the second step of our proposed model, we enforce the prediction of factors that have undergone changes. We assume that if some factor of two inputs is equal, then the variance of the corresponding latent vector dimension value is smaller compared to others. Based on this assumption, we define the target $(T)$ for factor prediction: if $\boldsymbol{z}_{1,i} - \boldsymbol{z}_{2,i} >$ threshold, then we set $T_i$ as 1 and 0 otherwise, where $T_i$ is a $i^{th}$ dimension value of $T \in \mathbb{R}^{|D|}$, $\boldsymbol{z}_{j,i}$ is an $i^{th}$ dimension value of $\boldsymbol{z}_j$, and we set the threshold as a hyper-parameter. For section prediction, we utilize the cross-entropy loss is defined as follows:

$$\mathcal{L}_p = \sum_{i=1}^{|S|} \sum_{c \in C} \mathbb{1}[T_i = c] \cdot \log P(T_i = c | [M; \Sigma]; [\boldsymbol{W}_s^i, \boldsymbol{b}_s^i]), \tag{7}$$

where $P(T_i = c | [M, \Sigma]; [\boldsymbol{W}_s^i, \boldsymbol{b}_s^i]) = p_s^i$, $p_s^i = [M; \Sigma]\boldsymbol{W}_s^i + \boldsymbol{b}_s^i$, $\boldsymbol{W}_s^i \in \mathbb{R}^{4|D| \times 2}$ and $\boldsymbol{b}_s^i \in \mathbb{R}^2$ are learnable parameters, and $c \in \{0, 1\}$.

To infer the activated section of the symmetries codebook, we utilize the Gumbel softmax function to handle binary on-and-off scenarios, akin to a switch operation:

$$sw(G(p_s^i)) = \begin{cases} G(p_{s,2}^i) & \text{if } p_{s,2}^i \geq 0.5 \\ 1 - G(p_{s,1}^i) & \text{if } p_{s,2}^i < 0.5 \end{cases}, \tag{8}$$

where $p_{s,j}^i$ is a $j^{th}$ dimension value of $p_s^i$, and $G(\cdot)$ is the Gumbel softmax with temperature as 1e-4.

**Integration for Composite Symmetry** For the composite symmetry $g_c$, we compute the product of weighted sums of switch function $sw(p_s)$ and prediction distribution $attn$ as:

$$g_c = \prod_{i=1}^{|S|} \hat{g}_c^i, \tag{9}$$

where $\hat{g}_c^i = \mathbf{e}^{sw(G(p_s^i)) \cdot \mathfrak{g}_c^i}$.

### 5.4 Equivariance Induction of Composite Symmetries

Motivated by the implementations of equivariant mapping in prior studies (Yang et al., 2022; Miyato et al., 2022) for disentanglement learning, we implement an equivariant encoder and decoder that satisfies $q_\phi(\psi_c * x_i) = g_c \circ q_\phi(x_i)$ and $p_\theta(g_c \circ z_i) = \psi_c * p_\theta(z_i)$ respectively, where $q_\phi$ is an encoder, $p_\theta$ is the decoder, $\psi_c * x_i = x_j$, and $g_c \circ z_i = z_j$. As shown in Fig. 2 example, we propose 1) the encoder equivariant loss $\mathcal{L}_{ee}$ to match the $z_i$ and $g_c \circ z_j$ on the latent vector space, and 2) the decoder equivalent loss $\mathcal{L}_{de}$ to match the input $x_i$ and generated image from the transformed vector $p_\theta(x|g_c \circ z_j)$.

**Equivariance Loss for Encoder and Decoder**  In the notation, $\psi_c$ and $g_i$ are group elements of the group $(\Psi, *)$ and $(\mathcal{G}, \circ)$ respectively, and both groups are isomorphic. Each group acts on the input and latent vector space with group action $*$, and $\circ$, respectively. We specify the form of symmetry $g_c$ and $\circ$ as an invertible matrix, and group action as matrix multiplication on the latent vector space. Then, we optimize the encoder and decoder equivariant function as:

$$g_c \circ q_\phi(x_i) = q_\phi(\psi_c * x_i) \iff g_c z_i - z_j \to 0$$
$$\Rightarrow \mathcal{L}_{ee} = \mathrm{MSE}(g_c z_i, z_j) \qquad \text{(for encoder).} \tag{10}$$

$$p_\theta(g_c \circ q_\phi(x_i)) = \psi_c * p_\theta(q_\phi(x_i)) \iff p_\theta(g_c \circ q_\phi(x_i)) - x_j \to 0$$
$$\Rightarrow \mathcal{L}_{de} = \mathrm{MSE}(p_\theta(g_c \circ q_\phi(x_i)), x_j) \text{ (for decoder).} \tag{11}$$

$$\mathcal{L}_{equiv} = \mathcal{L}_{ee} + \epsilon \mathcal{L}_{de} \tag{12}$$

For the equivariant encoder and decoder, we differently propose the single forward process by the encoder and decoder objective functions compared to previous work (Yang et al., 2022).

**Objective and Base model**  Our method can be plugged into existing VAE frameworks, where the objective function is integrated additively as follows:

$$\mathcal{L}(\phi, \theta; \boldsymbol{x}) = \mathcal{L}_{VAE} + \mathcal{L}_{codebook} + \mathcal{L}_{equiv}, \tag{13}$$

where $\mathcal{L}_{VAE}$ is the loss function of a VAE framework (Appendix A). The other loss $\mathcal{L}_{codebook} = \mathcal{L}_{pl} + \mathcal{L}_{pd} + \mathcal{L}_s + \mathcal{L}_c + \mathcal{L}_p$ and $\mathcal{L}_{equiv} = \mathcal{L}_{ee} + \epsilon \mathcal{L}_{de}$, where $\epsilon$ is a hyper-parameter to control the model performance sensitivity.

### 5.5 Extended Evaluation Metric: m-FVM Metric for Disentanglement in Multi-Factor Change

We define the *multi-factor change* condition as simultaneously altering more than two factors in the transformation between two samples or representations. To the best of our knowledge, there is no evaluation metric for disentanglement in multi-factor change, so we propose the extended version of the Factor-VAE metric (FVM) score called as multi-FVM score (m-FVM$_k$), where factor $F = F_1 \times \ldots \times F_n$, $k \in \{2, 3, \ldots, n-1\}$, and $|F_i|$ is a number of factors. Similar to FVM, 1) we randomly choose the factor $f = (f^{(i,\ldots,j)}, f^{-(i,\ldots,j)})$. 2) Then we fix the corresponding factor dimension value in the mini-batch. 3) Subsequently, we estimate the standard deviation (std.) of each dimension to find the number of $k$ lowest std. dimension $(z_{l1}, z_{l2}, \ldots)$ in one epoch. 4) We then count each pair of selected dimensions by std. values (the number of $(z_{l1}, z_{l2}, \ldots)$, which are corresponded to fixed factors). 5) In the last, we add the maximum value of the number of $(z_{l1}, z_{l2}, \ldots)$ on all fixed factor cases, and divide with epoch.

## 6 Experiments

**Device**  We set the below settings for all experiments in a single Galaxy 2080Ti GPU for 3D Cars and smallNORB, and a single Galaxy 3090 for dSprites 3D Shapes and CelebA. More details are in README.md file.

**Datasets**  The dSprites dataset Matthey et al. (2017) consists of 737,280 binary $64 \times 64$ images with five independent ground truth factors (number of values), *i.e.* x-position (32), y-position (32), orientation (40),

Table 3: Disentanglement scores for single factor change (left 5 metrics) and multi-factor change (m-FVMs) with 10 random seeds.

| 3D Car | FVM | beta VAE | MIG | SAP | DCI | m-FVM$_2$ | m-FVM$_3$ | m-FVM$_4$ |
|---|---|---|---|---|---|---|---|---|
| $\beta$-VAE | 91.83($\pm$4.39) | 100.00($\pm$0.00) | 11.44($\pm$1.07) | 0.63($\pm$0.24) | 27.65($\pm$2.50) | 61.28($\pm$9.40) | - | - |
| $\beta$-TCVAE | 92.32($\pm$3.38) | 100.00($\pm$0.00) | 17.19($\pm$3.06) | 1.13($\pm$0.37) | 33.63($\pm$3.27) | 59.25($\pm$5.63) | - | - |
| Factor-VAE | 93.22($\pm$2.86) | 100.00($\pm$0.00) | 10.84($\pm$0.93) | 1.35($\pm$0.48) | 24.31($\pm$2.30) | 50.43($\pm$10.65) | - | - |
| Control-VAE | 93.86($\pm$5.22) | 100.00($\pm$0.00) | 9.73($\pm$2.24) | 1.14($\pm$0.54) | 25.66($\pm$4.61) | 46.42($\pm$10.34) | - | - |
| CLG-VAE | 91.61($\pm$2.84) | 100.00($\pm$0.00) | 11.62($\pm$1.65) | 1.35($\pm$0.26) | 29.55($\pm$1.93) | 47.75($\pm$5.83) | - | - |
| CFASL | **95.70**($\pm$1.90) | **100.00**($\pm$0.00) | **18.58**($\pm$1.24) | **1.43**($\pm$0.18) | **34.81**($\pm$3.85) | **62.43**($\pm$8.08) | - | - |

| smallNORB | FVM | beta VAE | MIG | SAP | DCI | m-FVM$_2$ | m-FVM$_3$ | m-FVM$_4$ |
|---|---|---|---|---|---|---|---|---|
| $\beta$-VAE | 60.71($\pm$2.47) | 59.40($\pm$7.72) | 21.60($\pm$0.59) | 11.02($\pm$0.18) | 25.43($\pm$0.48) | 24.41($\pm$3.34) | 15.13($\pm$2.76) | - |
| $\beta$-TCVAE | 59.30($\pm$2.52) | 60.40($\pm$5.48) | 21.64($\pm$0.51) | 11.11($\pm$0.27) | 25.74($\pm$0.29) | 25.71($\pm$3.51) | 15.66($\pm$3.74) | - |
| Factor-VAE | 61.93($\pm$1.90) | 56.40($\pm$1.67) | **22.97**($\pm$0.49) | 11.21($\pm$0.44) | 24.84($\pm$0.72) | 26.43($\pm$3.47) | 17.25($\pm$3.50) | - |
| Control-VAE | 60.63($\pm$2.67) | 61.40($\pm$4.33) | 21.55($\pm$0.53) | 11.18($\pm$0.48) | **25.97**($\pm$0.43) | 24.11($\pm$3.41) | 16.12($\pm$2.53) | - |
| CLG-VAE | 62.27($\pm$1.71) | 62.60($\pm$5.17) | 21.39($\pm$0.67) | 10.71($\pm$0.33) | 22.95($\pm$0.62) | 27.71($\pm$3.45) | 17.16($\pm$3.12) | - |
| CFASL | **62.73**($\pm$3.96) | **63.00**($\pm$4.13) | 22.23($\pm$0.48) | **11.42**($\pm$0.48) | 24.58($\pm$0.51) | **27.96**($\pm$3.00) | **17.37**($\pm$2.33) | - |

| dSprites | FVM | beta VAE | MIG | SAP | DCI | m-FVM$_2$ | m-FVM$_3$ | m-FVM$_4$ |
|---|---|---|---|---|---|---|---|---|
| $\beta$-VAE | 73.54($\pm$6.47) | 83.20($\pm$7.07) | 13.19($\pm$4.48) | 5.69($\pm$1.98) | 21.49($\pm$6.30) | 53.80($\pm$10.29) | 50.13($\pm$11.98) | 48.02($\pm$8.98) |
| $\beta$-TCVAE | 79.19($\pm$5.87) | 89.20($\pm$4.73) | 23.95($\pm$10.13) | 7.20($\pm$0.66) | 35.33($\pm$9.07) | 61.75($\pm$6.71) | 57.82($\pm$5.39) | 53.71($\pm$4.22) |
| Factor-VAE | 78.10($\pm$4.45) | 84.40($\pm$5.55) | 25.74($\pm$10.58) | 6.37($\pm$1.82) | 32.30($\pm$9.47) | 58.39($\pm$5.18) | 51.63($\pm$2.88) | 53.71($\pm$4.22) |
| Control-VAE | 69.64($\pm$7.67) | 82.80($\pm$7.79) | 5.93($\pm$2.78) | 3.89($\pm$1.89) | 12.42($\pm$4.95) | 38.99($\pm$9.31) | 29.00($\pm$10.75) | 19.33($\pm$5.98) |
| CLG-VAE | **82.33**($\pm$5.59) | 86.80($\pm$3.43) | 23.96($\pm$6.08) | 7.07($\pm$0.86) | 31.23($\pm$5.32) | 63.21($\pm$8.13) | 48.68($\pm$9.59) | 51.00($\pm$8.13) |
| CFASL | 82.30($\pm$5.64) | **90.20**($\pm$5.53) | **33.62**($\pm$8.18) | **7.28**($\pm$0.63) | **46.52**($\pm$6.18) | **68.32**($\pm$0.13) | **66.25**($\pm$7.36) | **71.35**($\pm$12.08) |

| 3D Shapes | FVM | beta VAE | MIG | SAP | DCI | m-FVM$_2$ | m-FVM$_3$ | m-FVM$_4$ |
|---|---|---|---|---|---|---|---|---|
| $\beta$-VAE | 84.33($\pm$10.65) | 91.20($\pm$4.92) | 45.80($\pm$21.20) | 8.66($\pm$3.80) | 66.05($\pm$7.44) | 70.26($\pm$6.27) | 61.52($\pm$8.62) | 60.17($\pm$8.48) |
| $\beta$-TCVAE | 86.03($\pm$3.49) | 87.80($\pm$3.49) | 60.02($\pm$10.05) | 5.88($\pm$0.79) | 70.38($\pm$4.63) | 70.20($\pm$4.08) | 63.79($\pm$5.66) | **63.61**($\pm$5.90) |
| Factor-VAE | 79.54($\pm$10.72) | 95.33($\pm$5.01) | 52.68($\pm$22.87) | 6.20($\pm$2.15) | 61.37($\pm$12.46) | 66.93($\pm$17.49) | 63.55($\pm$18.02) | 57.00($\pm$21.36) |
| Control-VAE | 81.03($\pm$11.95) | 95.00($\pm$5.60) | 19.61($\pm$12.53) | 4.76($\pm$2.79) | 55.93($\pm$13.11) | 62.22($\pm$11.35) | 55.83($\pm$13.61) | 51.66($\pm$12.08) |
| CLG-VAE | 83.16($\pm$8.09) | 89.20($\pm$4.92) | 49.72($\pm$16.75) | 6.36($\pm$1.68) | 63.62($\pm$3.80) | 65.13($\pm$5.26) | 58.94($\pm$6.59) | 60.51($\pm$7.62) |
| CFASL | **89.70**($\pm$9.65) | **96.20**($\pm$4.85) | **62.12**($\pm$13.38) | **9.28**($\pm$1.92) | **75.49**($\pm$8.29) | **74.26**($\pm$2.82) | **67.68**($\pm$2.67) | 63.48($\pm$4.12) |

Table 4: Disentanglement performance rank. Each dataset rank is an average of evaluation metrics, and Avg. is an average of all datasets.

| | 3D Car | smallNORB | dSprites | 3D Shapes | Avg. |
|---|---|---|---|---|---|
| $\beta$-VAE | 3.33 | 4.86 | 4.88 | 3.38 | 4.11 |
| $\beta$-TCVAE | 2.50 | 4.29 | 2.50 | 2.88 | 3.04 |
| Factor-VAE | 3.17 | 2.71 | 3.38 | 4.00 | 3.31 |
| Control-VAE | 3.67 | 3.86 | 6.00 | 5.50 | 4.76 |
| CLG-VAE | 3.00 | 3.86 | 3.13 | 4.13 | 3.53 |
| CFASL | **1.00** | **1.43** | **1.13** | **1.13** | **1.17** |

shape (3), and scale (6). Any composite transformation of x- and y-position, orientation (2D rotation), scale, and shape is commutative. The 3D Cars Reed et al. (2015) dataset consists of 17,568 RGB $64 \times 64 \times 3$ images with three independent ground truth factors: elevations(4), azimuth directions(24), and car models(183). Any composite transformation of elevations(x-axis 3D rotation), azimuth directions (y-axis 3D rotation), and models are commutative. The smallNORB LeCun et al. (2004) dataset consists of total $96 \times 96$ 24,300 grayscale images with four factors, which are category(10), elevation(9), azimuth(18), light(6) and we resize the input as $64 \times 64$. Any composite transformation of elevations(x-axis 3D rotation), azimuth (y-axis 3D rotation), light, and category is commutative. 4) The 3D Shapes dataset Burgess & Kim (2018) consists of 480,000 RGB $64 \times 64 \times 3$ images with six independent ground truth factors: orientation(15), shape(4), floor color(10), scale(8), object color(10), and wall color(10). 5) The CelebA dataset Liu et al. (2015) consists of 202,599 images, and we crop the center $128 \times 128$ area and then, resize to $64 \times 64$ images.

**Evaluation Settings**  We set *prune_dims.threshold* as 0.06, 100 samples to evaluate global empirical variance in each dimension, and run it a total of 800 times to estimate the FVM score introduced in Kim & Mnih (2018). For the other metrics, we follow default values introduced in Michlo (2021), training and evaluation 10,000 and 5,000 times with 64 mini-batches, respectively Cao et al. (2022).

**Model Hyper-parameter Tuning**  We implement $\beta$-VAE Higgins et al. (2017), $\beta$-TCVAE Chen et al. (2018), control-VAE Shao et al. (2020), Commutative Lie Group VAE (CLG-VAE) Zhu et al. (2021), and Groupified-VAE (G-VAE) Yang et al. (2022) for baseline. For common settings to baselines, we set

Table 5: Comparison of disentanglement scores of plug-in methods in single factor change.

| Datasets | FVM | | MIG | | SAP | | DCI | |
|---|---|---|---|---|---|---|---|---|
| | G-VAE | CFASL | G-VAE | CFASL | G-VAE | CFASL | G-VAE | CFASL |
| dSprites | 69.75($\pm$13.66) | **82.30**($\pm$5.64) | 21.09($\pm$9.20) | **33.62**($\pm$8.18) | 5.45($\pm$2.25) | **7.28**($\pm$0.63) | 31.08($\pm$10.87) | **46.52**($\pm$6.18) |
| 3D Car | 92.34($\pm$2.96) | **95.70**($\pm$1.90) | 11.95($\pm$2.16) | **18.58**($\pm$1.24) | **2.10**($\pm$0.96) | 1.43($\pm$0.18) | 26.91($\pm$6.24) | **34.81**($\pm$3.85) |
| smallNROB | 46.64($\pm$1.45) | **61.15**($\pm$4.23) | 20.66($\pm$1.22) | **22.23**($\pm$0.48) | 10.37($\pm$0.51) | **11.12**($\pm$0.48) | **27.77**($\pm$0.68) | 24.59($\pm$0.51) |

batch size 64, learning rate 1e-4, and random seed from $\{1, 2, \ldots, 10\}$ without weight decay. We train for $3 \times 10^5$ iterations on dSprites smallNORB and 3D Cars, $6 \times 10^5$ iterations on 3D Shapes, and $10^6$ iterations on CelebA. We set hyper-parameter $\beta \in \{1.0, 2.0, 4.0, 6.0\}$ for $\beta$-VAE and $\beta$-TCVAE, fix the $\alpha, \gamma$ for $\beta$-TCVAE as 1 Chen et al. (2018). We follow the ControlVAE settings Shao et al. (2020), the desired value $C \in \{10.0, 12.0, 14.0, 16.0\}$, and fix the $K_p = 0.01$, $K_i = 0.001$. For CLG-VAE, we also follow the settings Zhu et al. (2021) as $\lambda_{hessian} = 40.0$, $\lambda_{decomp} = 20.0$, $p = 0.2$, and balancing parameter of $loss_{\text{rec group}} \in \{0.1, 0.2, 0.5, 0.7\}$. For G-VAE, we follow the official settings Yang et al. (2022) with $\beta$-TCVAE ($\beta \in \{10, 20, 30\}$), because applying this method to the $\beta$-TCVAE model usually shows higher performance than other models Yang et al. (2022). Then we select the best case of models. We run the proposed model on the $\beta$-VAE and $\beta$-TCVAE because these methods have no inductive bias to symmetries. We set the same hyper-parameters of baselines with $\epsilon \in \{0.1, 0.01\}$, threshold $\in \{0.2, 0.5\}$, $|S| = |SS| = |D|$, where $|D|$ is a latent vector dimension. More details for experimental settings.

## 6.1 Quantitative Analysis Results and Discussion

**Disentanglement Performance in Single and Multi-Factor Change** We evaluate four common disentanglement metrics: FVM (Kim & Mnih, 2018), MIG (Chen et al., 2018), SAP (Kumar et al., 2018), and DCI (Eastwood & Williams, 2018). As shown in Table 3, our method gradually improves the disentanglement learning in dSprites, 3D Cars, 3D Shapes, and smallNORB datasets in most metrics. To show the quantitative score of the disentanglement in multi-factor change, we evaluate the m-FVM$_k$, where $\max(k)$ is 2, 3, and 4 in 3D Cars, smallNORB, and dSprites datasets respectively. These results also show that our method positively affects single and multi-factor change conditions. As shown in Table 4, the proposed method shows a statistically significant improvement, as indicated by the higher average rank of dataset metrics compared to other approaches.

Table 6: Ablation study for loss functions on 3D-Cars and $\beta$-VAE with 10 random seeds.

| | $\mathcal{L}_p$ | $\mathcal{L}_c$ | $\mathcal{L}_{equiv}$ | $\mathcal{L}_{pl}$ | $\mathcal{L}_{pd}$ | $\mathcal{L}_s$ | FVM | MIG | SAP | DCI | m-FVM$_2$ |
|---|---|---|---|---|---|---|---|---|---|---|---|
| | ✗ | ✗ | ✗ | ✗ | ✗ | ✗ | 88.19($\pm$4.60) | 6.82($\pm$2.93) | 0.63($\pm$0.33) | 20.45($\pm$3.93) | 42.36($\pm$7.16) |
| | ✗ | ✓ | ✓ | ✓ | ✓ | ✓ | 88.57($\pm$6.68) | 7.18($\pm$2.52) | **1.85**($\pm$1.04) | 18.39($\pm$4.80) | 48.23($\pm$5.51) |
| | ✓ | ✓ | ✗ | ✓ | ✓ | ✓ | 88.56($\pm$7.78) | 7.27($\pm$4.16) | 1.31($\pm$0.70) | 19.58($\pm$4.45) | 42.63($\pm$4.21) |
| $\beta$-VAE | ✓ | ✓ | ✓ | ✗ | ✓ | ✓ | 86.95($\pm$5.96) | 7.11($\pm$3.49) | 1.09($\pm$0.40) | 18.35($\pm$3.32) | 41.90($\pm$7.80) |
| | ✓ | ✓ | ✓ | ✓ | ✗ | ✓ | 85.42($\pm$7.89) | 7.30($\pm$3.73) | 1.15($\pm$0.70) | 21.69($\pm$4.70) | 41.90($\pm$6.07) |
| | ✓ | ✓ | ✓ | ✗ | ✗ | ✗ | 89.34($\pm$5.18) | 9.44($\pm$2.91) | 1.26($\pm$0.40) | **23.14**($\pm$5.51) | 51.37($\pm$9.29) |
| | ✓ | ✓ | ✓ | ✓ | ✓ | ✗ | 90.71($\pm$5.75) | 9.29($\pm$3.74) | 1.07($\pm$0.65) | 22.74($\pm$5.06) | 45.84($\pm$7.71) |
| | ✓ | ✓ | ✓ | ✓ | ✓ | ✓ | **91.91**($\pm$3.45) | **9.51**($\pm$2.74) | 1.42($\pm$0.52) | 20.72($\pm$3.65) | **55.47**($\pm$10.09) |

**Comparison of Plug-in Methods** To compare our method with a wider range of approaches, we evaluate the disentanglement performance of the plug-in style method, G-VAE Yang et al. (2022) and apply both methods to $\beta$-TCVAE. As shown in Table 5, our method shows statistically significant improvements in disentanglement learning although $\beta$ hyper-parameter of CFASL is smaller than G-VAE.

**Ablation Study** Table 6 shows the ablation study to evaluate the impact of each component of our method for disentanglement learning. In the case of w/o $\mathcal{L}_{pl}$, the extraction of the composite symmetry $g_c$ becomes challenging due to the lack of unified roles among individual sections. Also, the coverage of code w/o $\mathcal{L}_{pd}$ is limited due to the absence of assurance that each section aligns with distinct factors. In the case of w/o $\mathcal{L}_s$, each section assigns a different role and the elements of each section align on the same factor, and w/o $\mathcal{L}_s$ case is better than w/o $\mathcal{L}_{pl}$ and w/o $\mathcal{L}_{pd}$. These results imply that the no-differentiated role of each section struggles with constructing adequate composite symmetry $g_c$. Also, it shows that dividing the symmetry information in each section ($\mathcal{L}_{pl}$, $\mathcal{L}_{pd}$) is more important than $\mathcal{L}_s$ for disentangled representation. To compare factor-aligned losses (w/o $\mathcal{L}_{pl}$, w/o $\mathcal{L}_{pd}$, w/o $\mathcal{L}_s$, and w/o $\mathcal{L}_{pl} + \mathcal{L}_{pd} + \mathcal{L}_s$), the best of among four cases is the w/o $\mathcal{L}_{pl} + \mathcal{L}_{pd} + \mathcal{L}_s$ and it implies that these losses are interrelated. Also, constructing the symmetries without the equivariant model is meaningless because the model does not satisfy Equation 10-12. The w/o $\mathcal{L}_{equiv}$ naturally shows the lowest results compared to other cases except w/o $\mathcal{L}_{pd}$ and $\mathcal{L}_{pl}$. Moreover, the w/o $\mathcal{L}_p$ case shows the impact of the section selection method (section 5.3) for unsupervised learning. Above all, each group exhibits a positive influence on disentanglement when compared to the base model ($\beta$-VAE). When combining all loss functions, our method consistently outperforms the others across the majority of evaluation metrics.

Table 7: Additional experiments

(a) Hyper-parameter tuning with 6 random seeds.

| $\epsilon$ | FVM | beta VAE | MIG | SAP | DCI |
|---|---|---|---|---|---|
| 0.01 | 76.98($\pm$8.63) | 87.33($\pm$7.87) | 29.68($\pm$11.38) | 6.96($\pm$1.16) | 41.28($\pm$11.93) |
| 0.1 | **82.21**($\pm$1.34) | **90.33**($\pm$5.85) | **34.79**($\pm$3.26) | **7.45**($\pm$0.61) | **48.07**($\pm$5.62) |
| 1.0 | 76.77($\pm$7.05) | 78.33($\pm$13.88) | 22.42($\pm$11.14) | 6.02($\pm$0.48) | 38.87($\pm$7.83) |

(b) Codebook size impact

| 3D Cars | $|\mathcal{G}|=100$ | $|\mathcal{G}|=10$ |
|---|---|---|
| FVM | **95.70**($\pm$1.90) | 48.63($\pm$24.55) |
| MIG | **18.58**($\pm$1.24) | 2.99($\pm$6.04) |
| SAP | **1.43**($\pm$0.18) | 0.29($\pm$0.34) |
| DCI | **34.81**($\pm$3.85) | 6.12($\pm$10.44) |
| FVM$_2$ | **62.43**($\pm$8.08) | 37.94($\pm$10.01) |

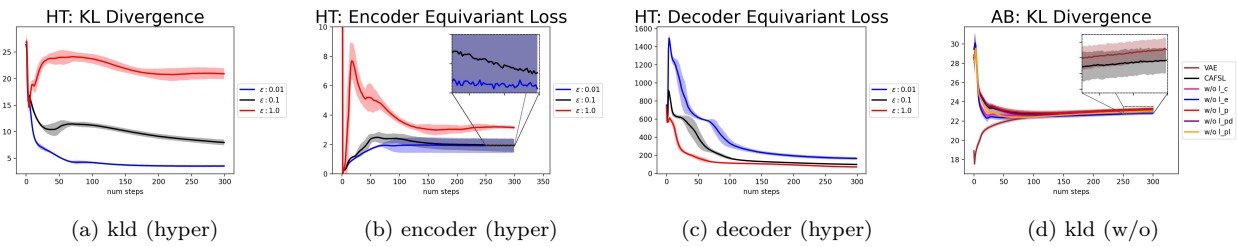

(a) kld (hyper)  (b) encoder (hyper)  (c) decoder (hyper)  (d) kld (w/o)

Figure 4: Loss curves: 1) HT: hyper-parameter tuning ($\epsilon \in \{0.01, 0.1, 1.0\}$) with $\beta$-TCVAE based CFASL. 2) AB: ablation study with $\beta$-VAE based CFASL.

**Impact of Hyper-Parameter Tuning** We operate a grid search of the hyper-parameter $\epsilon$. As shown in Figure 4a, the Kullback-Leibler divergence convergences to the highest value, when $\epsilon$ is large ($\epsilon = 1.0$) and it shows less stable results. It implies that the CFASL with larger $\epsilon$ struggles with disentanglement learning, as shown in Table 7a. Also, the $\mathcal{L}_{ee}$ in Figure 4b is larger than other cases, which implies that the model struggles with extracting adequate composite symmetry because its encoder is far from the equivariant model and it is also shown in Table 7a. Even though $\epsilon = 0.01$ case shows the lowest value in the most loss, $\mathcal{L}_{de}$ in Figure 4c is higher than others and it also implies the model struggles with learning symmetries, as shown in Table 7a because the model does not close to the equivariant model compare to $\epsilon = 0.1$ case.

**Posterior of CFASL** The symmetry codebook and composite symmetry are linear transformations of latent vectors. Intuitively, they enforce the posterior to be far from prior as $q_\phi(\boldsymbol{z}|\boldsymbol{x}) \sim \mathcal{N}(g_c\mu, g_c\Sigma g_c^\intercal)$, where $\mu$ and $\Sigma$ are close to zero vectors and identity matrix, respectively. However, as shown in Figure 4d, Kullbeck Leibler divergence is lower than VAE. It represents the ability of CFASL to preserve Gaussian normal distribution, which is similar to VAE.

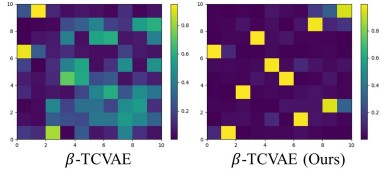

$\beta$-TCVAE  $\beta$-TCVAE (Ours)

Figure 5: Heatmaps of Eigenvectors for latent vector representations.

**Impact of Factor-Aligned Symmetry Size** We set the codebook size as 100, and 10 to validate the robustness of our method. In Table 7b, the larger size shows better results than the smaller one and is more stable by showing a low standard deviation.

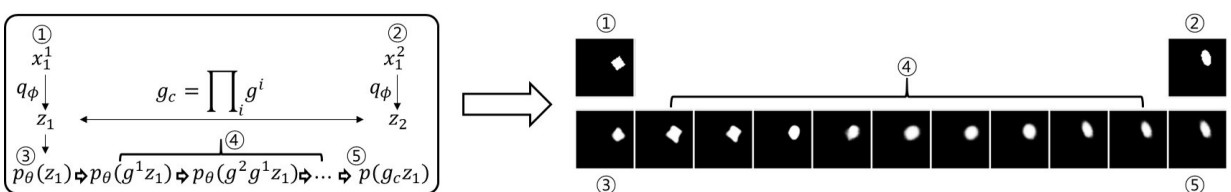

Figure 6: Generated images by composite symmetry and its factor-aligned symmetries. Image ① and ② are inputs, and image ③ is an output from image ① ($p_\theta(z_1)$). Image ⑤ is a output of group element $g_c$ acted on $z_1$ ($p(g_c z_1)$). Images ④ are outputs of decomposed composite symmetry $g_c$ acted on $z_1$ sequentially.

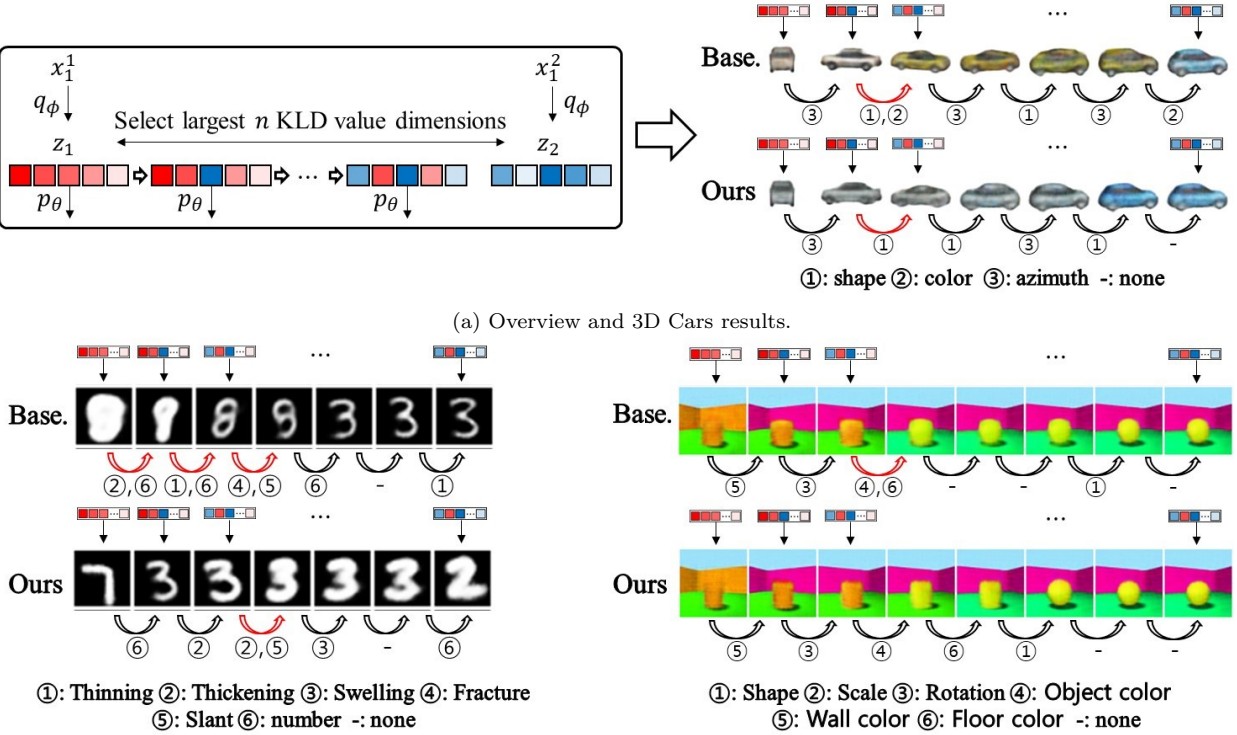

(a) Overview and 3D Cars results.

①: Thinning ②: Thickening ③: Swelling ④: Fracture
⑤: Slant ⑥: number  -: none

①: Shape ②: Scale ③: Rotation ④: Object color
⑤: Wall color ⑥: Floor color  -: none

(b) Morpho-MNIST and 3D Shapes results.

Figure 7: Generated images by dimension change. Red and blue colored squares represent the value of latent vector dimensions of $z_1$ and $z_2$. The images of baseline and CAFSL are the generated images from each latent vector.

## 6.2 Qualitative Analysis Results and Discussion

**Is Latent Vector Space Close to Disentangled Space?**   The previous result as shown in Figure 1 is a clear example of of whether the latent vector space closely approximates a disentangled space. The latent vector space of previous works (Figure 1a-1e) are far from disentangled space (Figure 1g) but CFASL shows the closest disentangled space compare to other methods.

**Alignment of Latent Dimensions to Factor-Aligned Symmetry**   In the principal component analysis of latent vectors shown in Figure 5, the eigenvectors $\boldsymbol{V} = [\boldsymbol{v}_1, \boldsymbol{v}_2, \ldots, \boldsymbol{v}_{|D|}]$ are close to one-hot vectors compared to the baseline, and the dominating dimension of the one-hot vectors are all different. This result implies that the representation (factor) changes are aligned to latent dimensions.

**Factor Aligned Symmetries**   To verify the representation of learnable codebook over composite symmetries and factor-aligned symmetries, we randomly select a sample pair as shown in Figure 6. The results imply that $g_c$ generated from the codebook represents the composite symmetries between two images (① and ②) because the image ② and the generated image ⑤ by symmetry $g_c$ are similar ($p(z_2) \approx p(g_c z_1)$). Also, each factor-aligned symmetry ($g_i$) generated from codebook section affects a single factor changes as shown in images ④. in Figure 6.

**Factor Aligned Latent Dimension**   To analyze each factor changes aligned to each dimension of latent vector space, we set the qualitative analysis as shown in Figure 7. We select the baseline and our model, which are the highest disentanglement performance scores. We select two random samples ($x_1$, $x_2$), generate latent vectors $z_1$ and $z_2$, and select the largest Kullback-Leibler divergence (KLD) value dimension from their posterior. Then, replacing the dimension value of $z_1$ to the value of $z_2$ one by one sequentially. As a result, the shape and color factors are changed when a single dimension value is replaced within the baseline

Figure 8: Generalization over unseen pairs of images. We set pairs $\{(x_{i-1}, x_i)|1 \leq i \leq ||\mathbb{X}|| - 1\}$ then extract the symmetries between elements of each pair $g_p = \{g_{(1,2)}, g_{(2,3)}, \dots g_{(k-1,k)}\}$ in inference step, where $g_{(k-1,k)}$ is a symmetry between $z_{k-1}$ and $z_k$. The first row images are inputs (targets) and the second row images are the generated images by symmetry codebook.

on the 3D Cars dataset, as shown in Figure 7a. However, our method results show no overlapped factor changes compared to baseline results. Also, the baseline results contain the changes of multiple factors in a dimension, but ours reduces the overlapped factors or contains only a single factor, as shown in Figure 7b. It shows that our model covers the diversity of datasets compared to the baseline.

**Unseen Change Prediction in Sequential Data**  The sequential observation as Miyato et al. (2022) is rarely observed in our methods, because of the random pairing during training (less 1 pair of observation). But their generated images via trained symmetries of our method are similar to target images as shown in Figure 8. This result implies that our method is strongly regularized for unseen change.

# 7 Conclusion

This work tackles the difficulty of disentanglement learning of VAEs in unknown factors change conditions. We propose a novel framework to learn composite symmetries from explicit factor-aligned symmetries by codebook to directly represent the multi-factor change of a pair of samples in unsupervised learning. The framework enhances disentanglement by learning an explicit symmetry codebook, injecting three inductive biases on the symmetries aligned to unknown factors, and inducing a group equivariant VAE model. We quantitatively evaluate disentanglement in the condition by a novel metric (m-FVM$_k$) extended from a common metric for a single factor change condition. This method significantly improved in the multi-factor change and gradually improved in the single factor change condition compared to state-of-the-art disentanglement methods of VAEs. Also, training process does not need the knowledge of factor information of datasets. This work can be easily plugged into VAEs and extends disentanglement to more general factor conditions of complex datasets.

# 8 Limitation and Future Work

In the real-world dataset, the variance of the factor is much more complex and has more combinations than the used datasets in this paper (the maximum number of factors is five). Although our method shows the advance of disentanglement learning in multi-factor change conditions, it remains the generalization in real world datasets or larger dataset. We consider extending the learning of composite symmetries in general conditions. Another drawback is to use of six loss functions, which require more hyper-parameter tuning. As the statistically learned group methods reduce hyper-parameters (Winter et al., 2022), we consider a more computationally efficient loss function.

**Acknowledgments**

This work was supported by the National Research Foundation of Korea (NRF) grant funded by the Korea government (MSIT) (No.2022R1A2C2012054, Development of AI for Canonicalized Expression of Trained Hypotheses by Resolving Ambiguity in Various Relation Levels of Representation Learning).

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

## A    Loss Function of Baseline

As shown in Table 8, we train the baselines with each objective function.

| VAEs | $\mathcal{L}_{VAE}$ |
|------|---------------------|
| $\beta$-VAE | $\mathbb{E}_{q_\phi(z\|x)} \log p_\theta(x\|z) - \beta \mathcal{D}_{KL}(q_\phi(z\|x)\|\|p(z))$ |
| $\beta$-TCVAE | $\mathbb{E}_{q_\phi(z\|x)} \log p_\theta(x\|z) - \alpha \mathcal{D}_{KL}(q(z,n)\|\|q(z)p(n))$ $-\beta \mathcal{D}_{KL}(q(z)\|\|\prod_j a(z_j)) - \gamma \sum_j \mathcal{D}_{KL}(q(z_j)\|\|p(z_j))$ |
| Factor-VAE | $\frac{1}{N} \sum_i^N [\mathbb{E}_{q(z\|x^i)}[\log p(x^i\|z)] - \mathcal{D}_{KL}(q(z\|x^i)\|\|p(z))]$ $-\gamma \mathcal{D}_{KL}(q(z)\|\|\prod_j(z_j))$ |
| Control-VAE | $\mathbb{E}_{q_\phi(z\|x)} \log p_\theta(x\|z) - \beta(t)\mathcal{D}_{KL}(q_\phi(z\|x)\|\|p(z))$ |
| CLG-VAE | $\mathbb{E}_{a(z\|x)q(t\|z)} \log p(x\|z)p(z\|t)$ $-\mathbb{E}_{q(z\|x)}\mathcal{D}_{KL}(q(t\|z)\|\|p(t)) - \mathbb{E}_{q(z\|x)} \log q(z\|x)$ |

Table 8: Objective Function of the VAEs.

## B    Perpendicular and Parallel Loss Relationship

We define parallel loss $\mathcal{L}_p$ to set two vectors in the same section of the symmetries codebook to be parallel: $z - g_j^i \parallel z - g_{j'}^i z$ then,

$$z - g_j^i z = c(z - g_{j'}^i z) \tag{14}$$

$$\Rightarrow (1-c)z = (g_j^i - cg_{j'}^i)z \tag{15}$$

$$\Rightarrow (1-c)I = g_j^i - cg_{j'}^i \text{ or } [(1-c)I + cg_{j'}^i - g_j^i]z = 0, \tag{16}$$

where $I$ is an identity matrix and constant $c \in \mathbb{R}$. However, all latent $z$ is not eigenvector of $[(1-c)I + cg_{j'}^i - g_j^i]$. Then, we generally define symmetry as:

$$g_{j'}^i = \frac{1}{c}g_j^i + \frac{c-1}{c}I, \tag{17}$$

where $i, j$, and $j'$ are natural number $1 \le i \le |S|$, $1 \le j, j' \le |SS|$, and $k \ne j$. Therefore, all symmetries in the same section are parallel then, any symmetry in the same section is defined by a specific symmetry in the same section.

We define orthogonal loss $\mathcal{L}_o$ between two vectors, which are in different sections, to be orthogonal: $z - g_j^i z \perp z - g_l^k z$, where $i \ne k$, $1 \le i, k \le |S|$, and $1 \le j, l \le |SS|$. By the Equation 17,

$$z - g_j^i z \perp z - g_l^k z \tag{18}$$

$$\Rightarrow (\frac{1}{c_a}g_a^i + \frac{c_a-1}{c_a}I)z - z \perp (\frac{1}{c_b}g_b^k + \frac{c_b-1}{c_b}I)z - z \tag{19}$$

$$\Rightarrow \frac{1}{c_a}(g_a^i z - z) \perp \frac{1}{c_b}(g_b^k z - z), \tag{20}$$

where $c_a$ and $c_b$ are any natural number, and $1 \le a, b \le |SS|$. Therefore, if two vectors from different sections are orthogonal and satisfied with Equation 17, then any pair of vectors from different sections is always orthogonal.

## C    Additional Experiment

### C.1    Disentanglement Performance

**Reconstruction vs. FVM**   We conduct the trade-off between the reconstruction and disentanglement performance as shown in Figure 9. We consider the results on the complex dataset because the trade-off is more distinct in the complex dataset such as 3DShapes.

**Complex Dataset**   We show the model performance with MPI3D dataset.

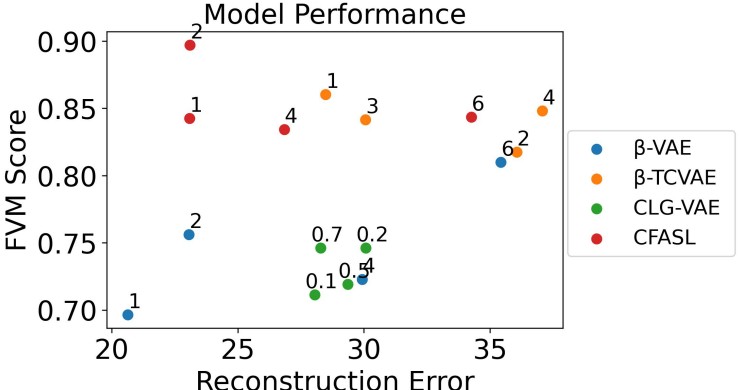

Figure 9: Reconstruction loss vs. Factor VAE metric on 3D Shapes dataset. The numbers next to each plot represent the value of $loss_{\text{rec group}}$ of CLG-VAE and the others are the value of $\beta$ parameter.

|  | FVM | beta VAE |
|---|---|---|
| $\beta$-VAE | 16.79($\pm$0.80) | 36.40($\pm$8.53) |
| $\beta$-TCVAE | 16.95($\pm$ 1.38) | 51.75($\pm$9.22) |
| CFASL | **17.44**($\pm$3.33) | **54.80**($\pm$4.54) |

(a) Disentanglement performance on the MPI3D dataset.

| $p$-value | FVM | MIG | SAP | DCI |
|---|---|---|---|---|
| dSprites | **0.011** | **0.005** | **0.016** | **0.001** |
| 3D Cars | **0.006** | **0.000** | 0.97 | **0.003** |
| smallNORB | **0.000** | **0.002** | **0.000** | 1.000 |

(b) $p$-value estimation on each datasets.

| 3D Shapes | $\beta$-VAE | $\beta$-TCVAE | Factor-VAE | Control-VAE | CLG-VAE | OURS |
|---|---|---|---|---|---|---|
| m-FVM$_5$ | 80.26($\pm$3.78) | 79.21($\pm$5.87) | 76.69($\pm$5.08) | 73.31($\pm$6.54) | 73.61($\pm$4.22) | **83.03**($\pm$2.73) |

(c) m-FVMs results.

Table 9: Additional experiments

**Statistically Significant Improvements** As shown in Table 9b, our model significantly improves disentanglement learning.

**3D Shapes** As shown in Table 9c, CFASL also shows an advantage on multi-factor change.

### C.2 Ablation Studies

**How Commutative Lie Group Improves Disetanglement Learning?** The Lie group is not commutative, however most factors of the used datasets are commutative. For example, 3D Shapes dataset factors consist of the azimuth (x-axis), yaw (z-axis), coloring, scale, and shape. Their 3D rotations are all commutative. Also, other composite symmetries as coloring and scale are commutative. Even though we restrict the Lie group to be commutative, our model shows better results than baselines as shown in Table 3.

**Impact of Commutative Loss on Computational Complexity** As shown in Table 10, our methods reduce the composite symmetries computation. Matrix exponential is based on the Taylor series and it needs high computation

| 3D Cars | $\mathcal{L}_c$ | without $\mathcal{L}_c$ |
|---|---|---|
|  | x**4.63** | x1.00 |

Table 10: Complexity.

cost though its approximation is lighter than the Taylor series. We need one matrix exponential computation for composite symmetries with commutative loss, in contrast, the other case needs the number of symmetry codebook elements $|S| \cdot |SS|$ for the matrix exponential and also $|S| \cdot |SS| - 1$ time matrix multiplication.

### C.3    Additional Qualitative Analysis (Baseline vs. CFASL)

Figure 10 and 14a show the qualitative results on 3D Cars introduced in Figure 6-8. Figure 11, and Figure 13 show the dSprites and smallNORB dataset results, respectively. Additionally, we describe Figure 12 and 14c results over 3D Shapes datasets. We randomly sample the images in all cases.

**3D Cars**    As shown in Figure 10c, CFASL shows better results than the baseline. In the $1^{st}$ and $2^{nd}$ rows, the baseline changes shape and color factor when a single dimension value is changed, but ours clearly disentangle the representations. Also in the $3^{rd}$ row, the baseline struggles with separating color and azimuth but CFASL successfully separates the color and azimuth factors.

- $1^{st}$ row: our model disentangles the *shape* and *color* factors when the $2^{nd}$ dimension value is changed.

- $2^{nd}$ row: ours disentangles *shape* and *color* factors when the $1^{st}$ dimension value is changed.

- $4^{th}$ row: ours disentangles the *color*, and *azimuth* factors when the $2^{nd}$ dimension value is changed.

**dSprites**    As shown in Figure 11c, the CFASL shows better results than the baseline. The CFASL significantly improves the disentanglement learning as shown in the $4^{th}$ and $5^{th}$ rows. The baseline shows the multi-factor changes during a single dimension value is changed, while ours disentangles all factors.

- $1^{st}$ row: ours disentangles *the x- and y-pos* factor when the $2^{nd}$ dimension value is changed.

- $2^{nd}$ row: ours disentangles the *rotation* and *scale* factor when the $2^{nd}$ dimension value is changed.

- $3^{rd}$ row: ours disentangles the *x- and y-pos*, and *rotation* factor when the $1^{st}$ and $2^{nd}$ dimension values are changed.

- $4^{th}$ row: ours disentangles the *all factors* when the $1^{st}$ and $2^{nd}$ dimension values are changed.

**3D Shapes**    As shown in Figure 12c, the CFASL shows better results than the baseline. In the $1^{st}$, $3^{rd}$, and $5^{th}$ rows, our model clearly disentangles the factors while the baseline struggles with disentangling multi-factors. Even though our model does not clearly disentangle the factors, compared to the baseline, which is too poor for disentanglement learning, ours improves the performance.

- $1^{st}$ row: our model disentangles the *object color* and *floor color* factor when the $2^{nd}$ and $3^{rd}$ dimension values are changed.

- $2^{nd}$ row: ours disentangles *shape* factor in $1^{st}$ dimension, and *object color* and *floor color* factors at the $4^{th}$ dimension value are changed.

- $3^{rd}$ row: ours disentangles the *object color* and *floor color* factor when the $3^{rd}$ dimension value is changed.

- $4^{th}$ row: ours disentangles the *scale*, *object color*, *wall color*, and *floor color* factor when the $2^{nd}$ and $3^{rd}$ dimension values are changed.

- $5^{th}$ row: ours disentangles the *shape*, *object color*, and *floor color* factor when the $1^{st}$ and $2^{nd}$ dimension values are changed.

**smallNORB**    Even though our model does not clearly disentangle the multi-factor changes, ours shows better results than the baseline as shown in Figure 13c.

- $1^{st}$ row: our model disentangles the *category* and *light* factor when the $2^{nd}$ dimension value is changed.

- $3^{rd}$ row: ours disentangles *category* factor and *azimuth* factors when the $5^{th}$ dimension value is changed.

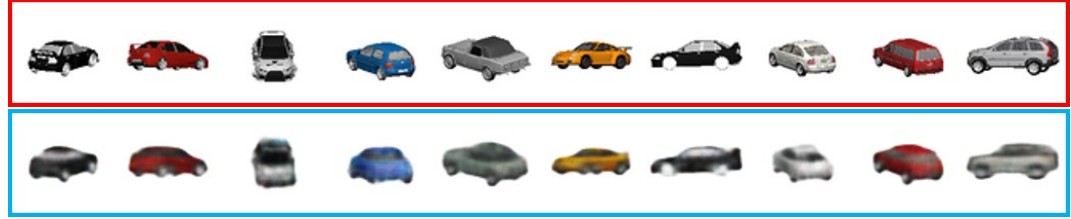

(a) Generated images by composite symmetry on 3D Cars dataset. The images in the red box are inputs. The images in the blue box at odd column are same as ③ and even column are same as ⑤ in Figure 6.

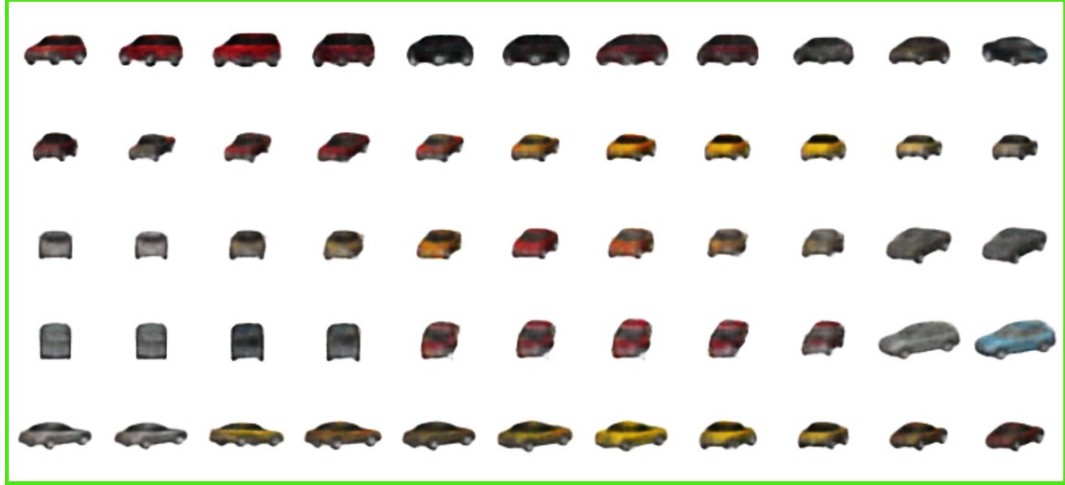

(b) Generated images by its factor-aligned symmetries on 3D Cars dataset. The images are same as ④ in Fig 6.

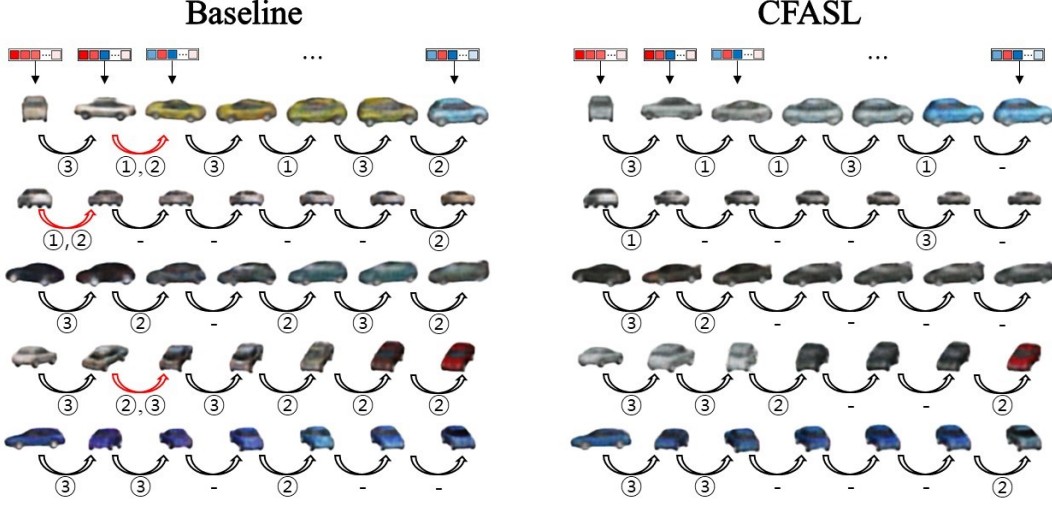

(c) Generated images by dimension change on 3D Cars dataset.

Figure 10: Figure 10a shows the generation quality of composite symmetries results, Figure 10b shows the disentanglement of symmetries by factors results, and Figure 10c shows the disentanglement of latent dimensions by factors results.

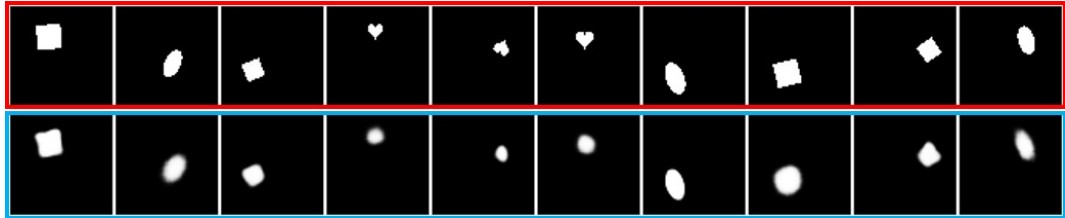

(a) Generated images by composite symmetry on dSprites dataset. The images in the red box are inputs. The images in the blue box at odd column are same as ③ and even column are same as ⑤ in Figure 6.

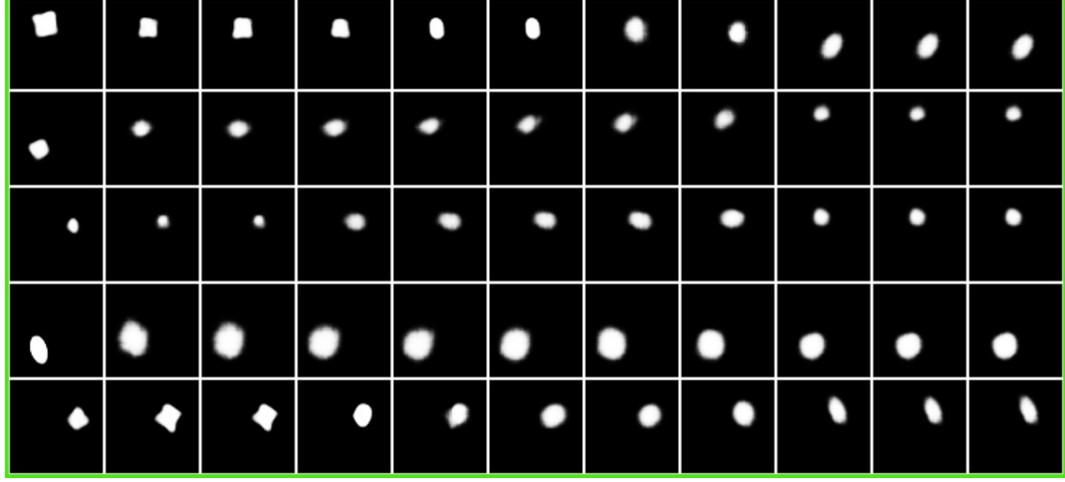

(b) Generated images by its factor-aligned symmetries on dSprites datset. The images are same as ④ in Fig 6.

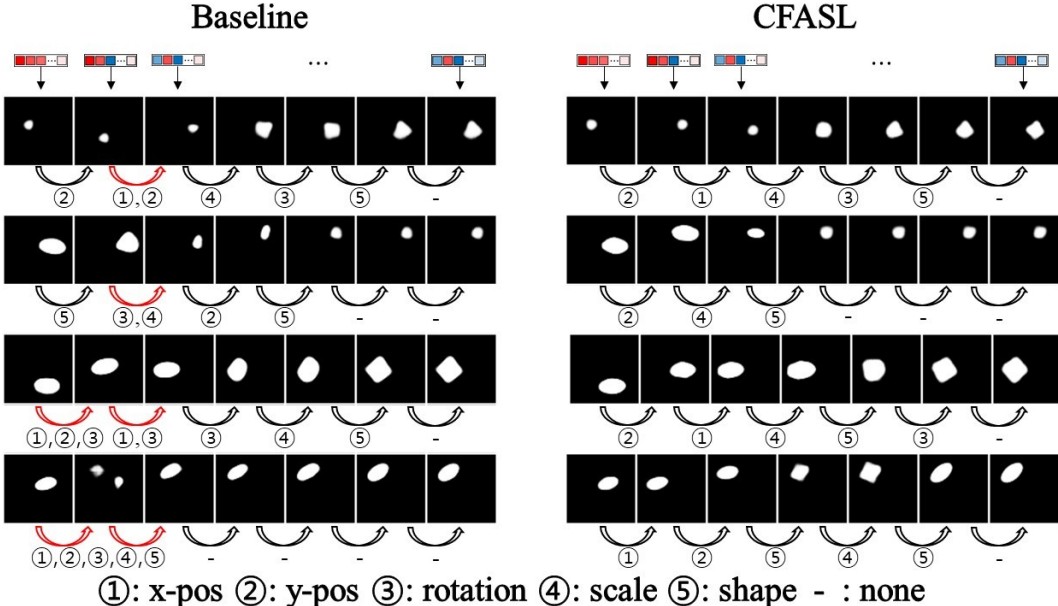

(c) Generated images by dimension change on dSprites dataset.

Figure 11: Figure 11a shows the generation quality of composite symmetries results, Figure 11b shows the disentanglement of symmetries by factors results, and Figure 11c shows the disentanglement of latent dimensions by factors results.

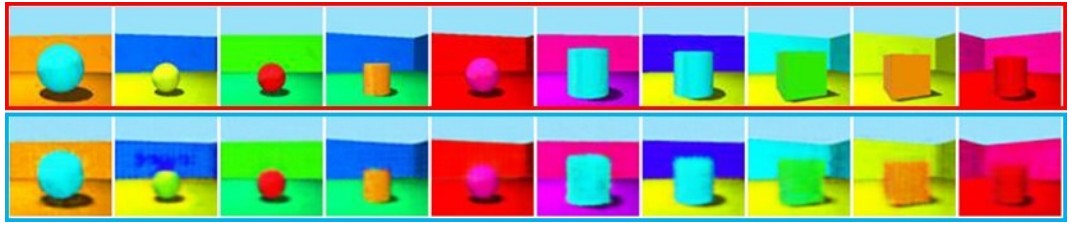

(a) Generated images by composite symmetry on 3DShapes dataset. The images in the red box are inputs. The images in the blue box at odd column are same as ③ and even column are same as ⑤ in Figure 6.

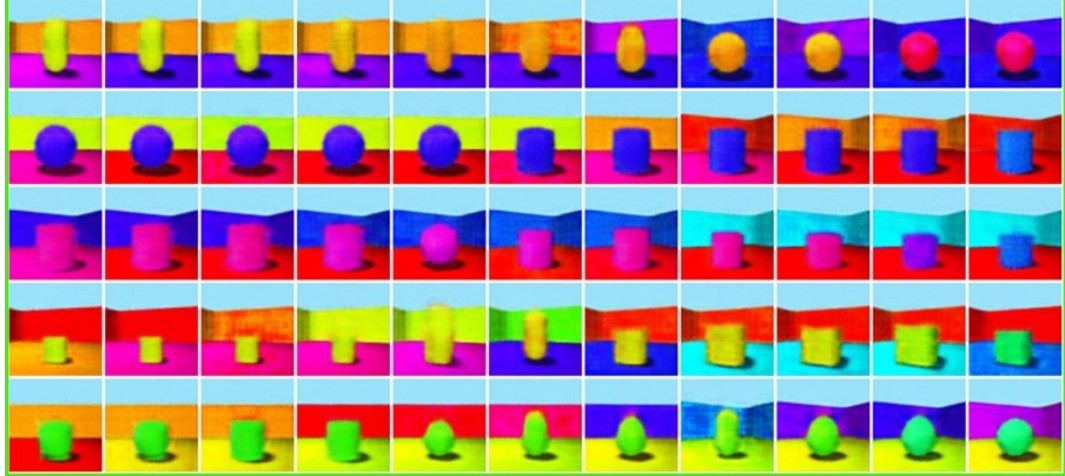

(b) Generated images by its factor-aligned symmetries on 3DShapes datset. The images are same as ④ in Fig 6

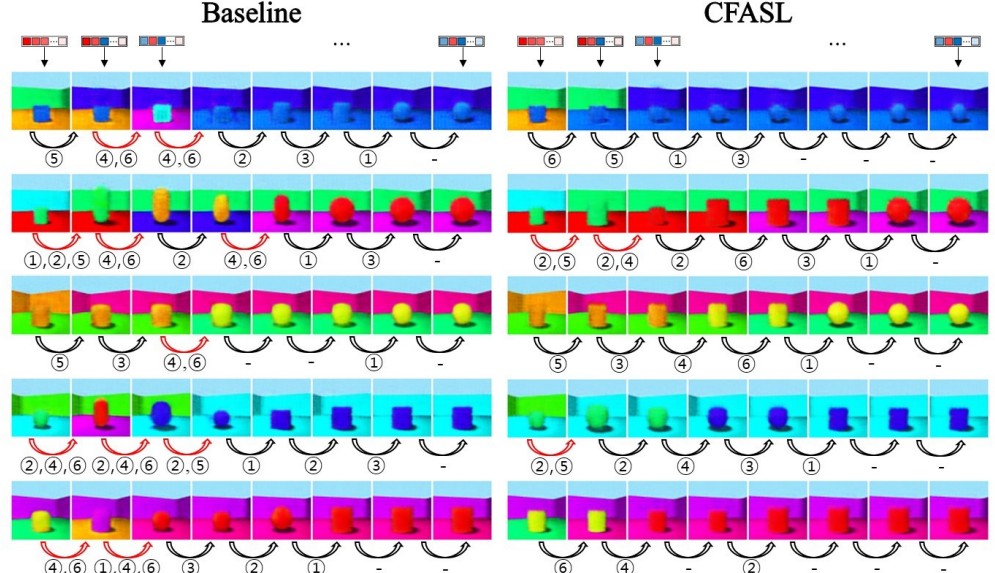

①: shape ②: scale ③: rotation ④: object color ⑤: wall color ⑥: floor color - : none

(c) Generated images by dimension change on 3DShapes dataset.

Figure 12: Figure 12a shows the generation quality of composite symmetries results, Figure 12b shows the disentanglement of symmetries by factors results, and Figure 12c shows the disentanglement of latent dimensions by factors results.

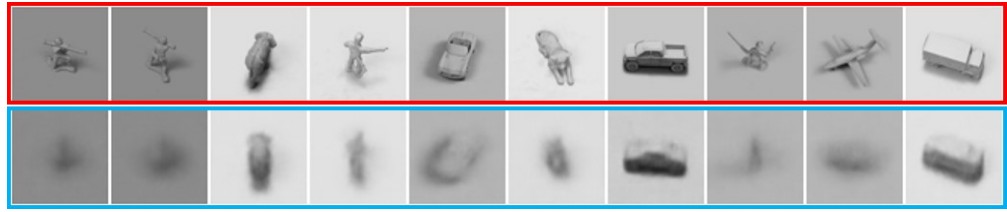

(a) Generated images by composite symmetry on smallNORB dataset. The images in the red box are inputs. The images in the blue box at odd column are same as ③ and even column are same as ⑤ in Figure 6.

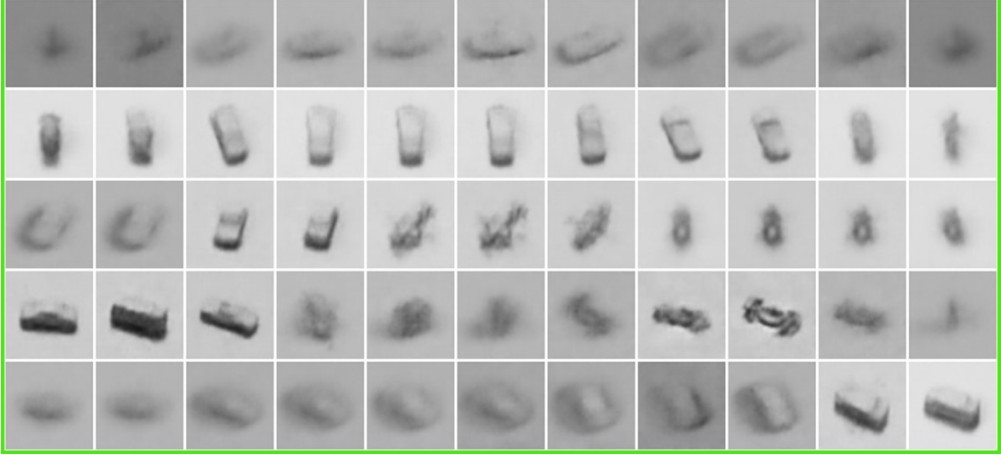

(b) Generated images by its factor-aligned symmetries on smallNORB datset. The images are same as ④ in Fig 6

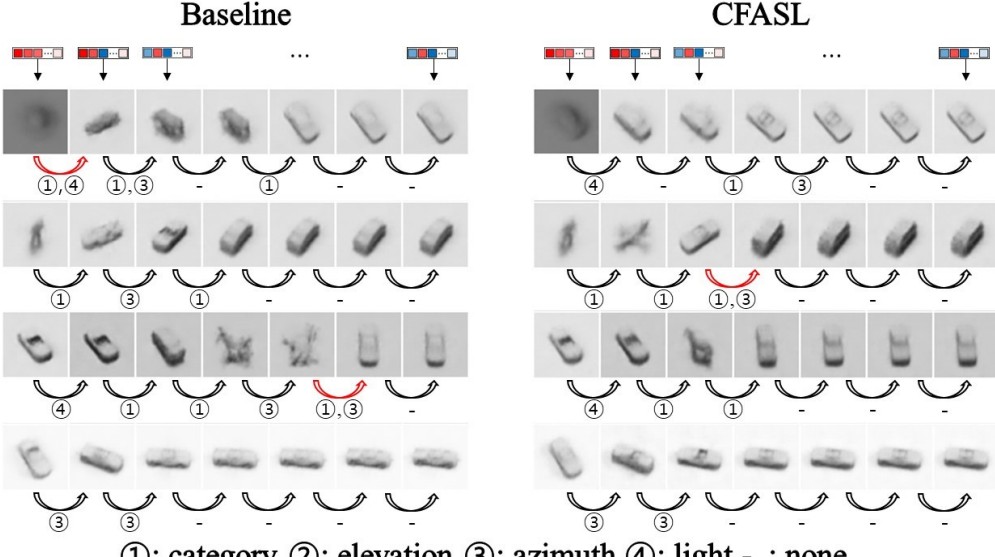

(c) Generated images by dimension change on smallNORB dataset.

Figure 13: Figure 13a shows the generation quality of composite symmetries results, Figure 13b shows the disentanglement of symmetries by factors results, and Figure 13c shows the disentanglement of latent dimensions by factors results.

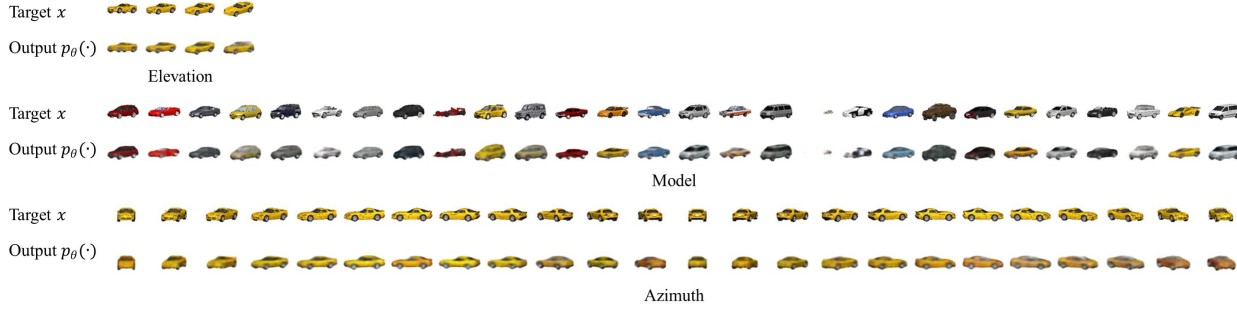

(a) Generalization over unseen pairs of images on 3D Car dataset.

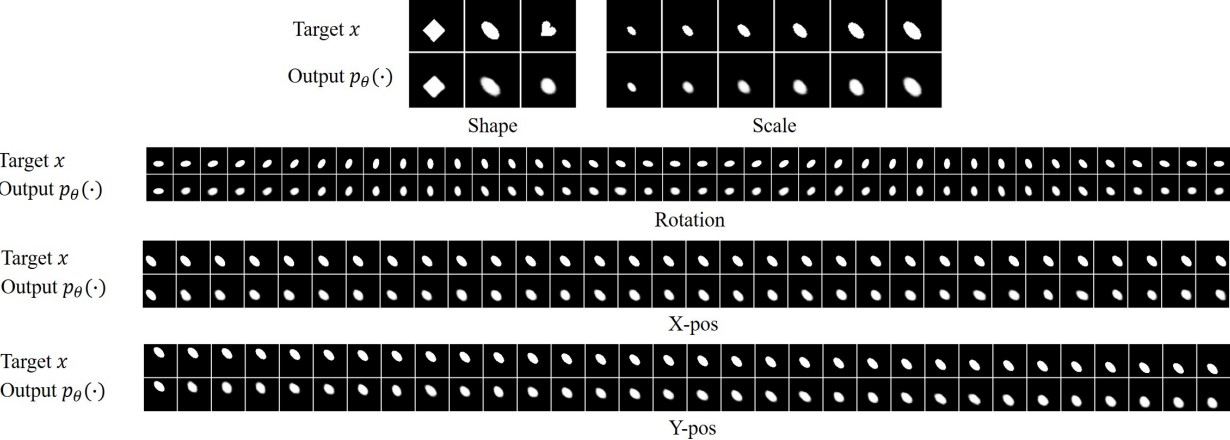

(b) Generalization over unseen pairs of images on dSprites dataset.

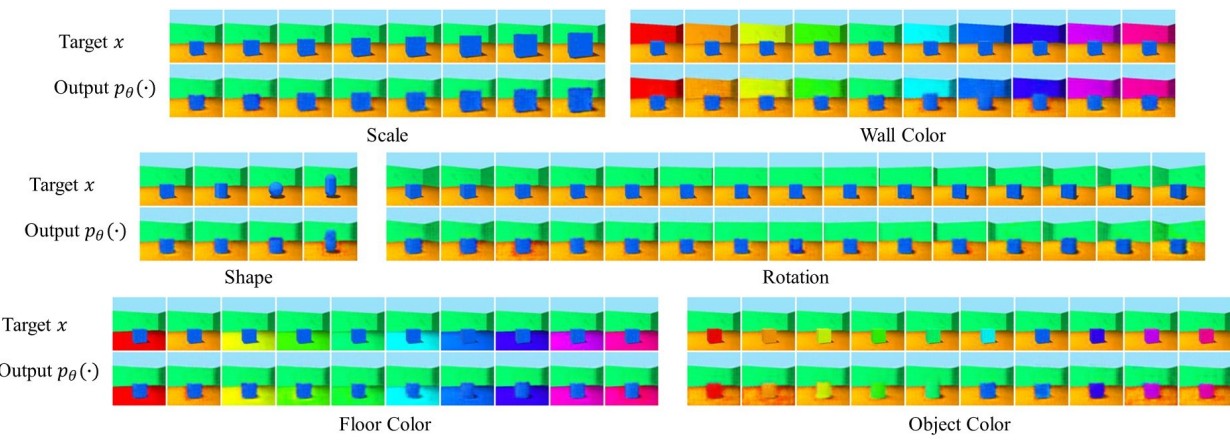

(c) Generalization over unseen pairs of images on 3DShapes dataset.

Figure 14: Generalization over unseen pairs of images. The proposed method represents the sequential symmetries over the whole factor of each dataset. Generated images follow the same process of Figure 8.

