# OpenReview forum: "CFASL: Composite Factor-Aligned Symmetry Learning for Disentanglement in Variational AutoEncoder"
_TMLR — Accepted by TMLR_

### Review · Reviewer_33dK · 2024-08-21

**Summary Of Contributions:**

The paper titled proposes a new approach, dubbed CFASL, for unsupervised symmetry-based disentanglement in VAEs without requiring prior knowledge of dataset factors. The method integrates three key components:

(i) Inductive bias for latent alignment: CFASL aligns latent vector dimensions with factor-aligned symmetries using a learnable symmetry codebook, which facilitates better disentanglement.
(ii) Composite symmetry learning: Expresses changes between random samples by learning factor-aligned symmetries that capture complex transformations within the data.
(iii) Group equivariant encoder and decoder: The model employs a group equivariant encoder and decoder to reinforce these symmetries during VAE training.

The paper then provides a rather broad empirical study that demonstrates the performance of CFASL on several benchmark datasets, comparing it with existing state-of-the-art methods.

**Audience:**

Yes

**Broader Impact Concerns:**

No broader impact concerns.

**Claims And Evidence:**

Yes

**Requested Changes:**

I request that the authors address the weaknesses described above according to the provided suggestions.

**Strengths And Weaknesses:**

**Strengths:** I find the considered task (namely, learning a factor aligned VAE whose latent space holds in correspondence to symmetries in the data) very interesting. While I have some reservations regarding the proposed approach, I view the following as strengths of this work:

1. The proposed method has several new components on top of existing approaches, which enable extracting transformations between inputs and defining symmetries on the latent space. This is achieved using an explicit codebook for symmetries and an injected inductive bias that matches each symmetry to a single factor change.

2. The methodology is decomposed into steps, for which detailed explanations are provided across the subsections of Section 4. The reviewer appreciates the idea behind this presentation approach, which aims to split the overall cumbersome idea into smaller digestible pieces.

3. The empirical performance is overall compelling, with CFASL presenting a clear improvement over competing approaches in many of the tested scenarios.

4. Although not viewed as a major contribution, the extended evaluation metric to assess multi-factor changes proposed in this work is noted. This seems like a useful idea that may be employed for empirical evaluation in follow-up works.

**Weaknesses:** While, as mentioned above, the considered task is interesting and important, I would like to start from a general comment on the state of the factor aligned VAE literature. The current approaches, including the one proposed in this work, are quite reductionist. They stitch up a model and loss from various components that are separately taking care of different aspects of the problem---in this case factor alignment for symmetries. A holistic and more interpretable approach that automatically adapts to those symmetries will be the ultimate solution, but I understand that this literature is not there yet. This is not held as criticism of the current submission. Rather, an observation of what I believe is an unsatisfactory aspect of current methodologies.

Beyond that, given that this is a heuristic methodology paper, clarity in presentation and a valid justification of the proposed steps are crucial. However, as outlined below, I find that the current submission falls short in these areas:

1.  The preliminaries section is massively lacking in background. Very basic ideas from group theory are presented, but central ideas for the proposed approach are not covered, making subsequent explanations hard to understand. This includes, for instance, background on the $\beta$-VAE framework, specification of input/output/latent spaces, structural assumptions on the data (e.g., the symmetries it possesses), Lie groups/algebras, factor-VAE metric, etc. A notation section would also be good to add. In its current state, the paper is not stand-alone and will therefore be difficult to read for any researcher not directly in the field of study.

2. The paper significantly lacks in the level of presentation. This is expressed not only in multiple typos, capitalization, and grammatical errors (e.g., "onn" above Fig.1, "[...]. we" and "lie group" in the paragraph below Fig. 2, "not worth" before Eq. 3, $||$ symbol before Eq. (1) in not defined, the first sentence in the last paragraph on page 7 start from half a sentence, broken indicator symbol in Eq. (4), and many others), but also in the explanations of the various steps in Section 4 being hard to follow. These explanations are dry and do not provide intuition for what is being done. They are also full of jargon, undefined notation, and typos, which makes then even harder to follow. This hampers the reproducibility and the rationale behind the proposed approach.
First of all, the text requires significant polishing and a few rounds of English editing. Beyond that, I believe that incorporating a simple running example (e.g., a 2D image with rotations) to support these explanations would be helpful. I encourage the authors to consider doing so, and refer back to this example in each subsection, crystalizing each step and the relevant mathematical expressions in its context.

3. Figure 2 is supposed to summarize the method and provide a comprehensive view. However, I think that it fails to do so because it is hard to parse, with neither the caption nor the text providing sufficient information to make sense of it. I suggest breaking it up into labeled subfigures and expanding the caption (or the relevant parts of the text) to provide a clearer description that addresses both the intuition and the mathematical expressions/formulae. For instance, in Section 4.3 it says: "In the first step, the model generates the factor aligned symmetries of each section through the attention score as shown in Fig. 2", but it is not explained which part of Fig. 2 the reader should refer to or what they should look for. This shortcoming repeats itself throughout the text and applies to other figures as well (often referred to by merely saying `as shown in Fig. x', without further context).

4. It is unclear why there is a hyperparameter $\epsilon$ weighting $L_{ee}$ and $L_{de}$ in $L_{equiv}$, but no such parameters are introduced within $L_{codebook}$. The authors should clarify this point and justify this design choice.

5. The contribution in defining m-FVM$_k$ is hard to appreciate without proper background being provided for the vanilla FVM.

6. Section 5 on related work seems misplaced and should appear as part of the introduction. If the goal is to review the competing methods before presenting the experiments, then the authors should include a dedicated paragraph summarizing the specific approach that CFASL will be compared to.

Overall, I think that the paper presents a promising method with evident practical advantages. However, clarity of presentation and reproducibility are key aspects in which I think that the current submission significantly lacks.

---

> ### Author Response · Authors · 2024-09-19
> **Response to Reviewer 33dK by Authors (1)**
>
> Dear reviewer 33dK,
>
> We appreciate your valuable and fruitful comments to improve this research. The abbreviation ‘RW’ refers to a reviewer, ‘RQ’ refers to a request, and ‘A’ refers to an answer.
>
> ---
>
> RW1. RQ. 1. The preliminaries section is massively lacking in background. This includes, for instance, background on 1) Lie groups/algebras 2) the β-VAE framework, 3) factor-VAE metric, 4) specification of input/output/latent spaces, structural assumptions on the data (e.g., the symmetries it possesses), , etc.
> >A1.
> >
> >1.	We added Lie group and Lie algebra background on Section 2.1, in the paragraph ‘Lie group and Lie algebra’ in revised version.
> >2.	We added β-VAE framework on Section 2.2, in the paragraph ‘Variational Auto-Encoder(VAE)’ in revised version.
> >3.	We added factor-VAE metric on Section 2.2, in the paragraph ‘Factor VAE Metric (FVM)’ in revised version.
> >4.	We add specification of input/output/latent spaces, structural assumptions on the data in Section 2.3, in the paragraph ‘Disentangled Representation Based on the Group’ in revised version.
> We highlighted these in the revised paper.
>
> ---
>
> RW1. RQ 2. A notation section would also be good to add.
> >A2. We added the notation table (Table 1 in revised version), and we highlighted it.
>
> ---
>
> RW1. RQ 3. This is expressed not only in multiple typos, capitalization, and grammatical errors
> >A3.
> >
> >1. We modified the typo and grammar.
> 2. The symbol || represents parallelism, and we added this symbol in the notation table (Table 1. in the revised paper)
> 3. We modified this sentence: “However, it is not worthy that neither the parallel loss nor the perpendicular loss exhibits specific constraints aimed at inducing this characteristic as shown in Fig.” to “However, the parallel and the perpendicular loss do not factorize latent dimension as shown in Figure 3.”
>
> ---
>
>
> RW1. RQ 4. First of all, the text requires significant polishing and a few rounds of English editing. Beyond that, I believe that incorporating a simple running example (e.g., a 2D image with rotations) to support these explanations would be helpful.
> >A4. We added the example to illustrate the entire model process on caption of Figure 2 in the revised paper. Our model process is as follows:
> >
> >1. A pair of images (e.g., $differences$ between two images are in the x- and y-position) is given, and the goal of the model is to represent the differences on the latent vector space, called the composite symmetry $g_c$ for disentangled representations.
> >2. The $codebook$ is designed to represent the composite symmetry $g_c$.
> >3. Each section of the codebook is separated to affect a single factor e.g., the $i^{th}$ section affects the x-position, and the $j^{th}$ section affects the y-position of images.
> >4. Each section consists of Lie algebra to provide diversity of symmetries.
> >5. Each loss optimizes the codebook to guarantee the 3) as follows:
> >    1. Symmetries from the same section affect the same factor e.g., $exp(\mathfrak{g}^i_k)$, where $k \in \\{1, 2, \ldots, |SS|\\}$
> >    2. Each section affects different factors, e.g., $g_c^1, g_c^2, \ldots$ affect x-position, y-position, and, etc.
> >    3. Each section changes a single dimension of latent vectors for disentangled representation.
> >6. $Attn$ ensures diversity in symmetries representation, and $p_s$ predicts the activate dsection, int he case, $i^{th}$ and $j^{th}$ sections for x- and y-position differences.
> >7. The model then represents the composite symmetry $g_c$.
> >8. Lastly, model optimizes the $L_{ee}$ to match $g_c z_1 (=z_2^\prime)$, and $L_{de}$ to match the $x_1$ and $p_\theta(g_c \circ q_\phi(x_2))$ to inject the inductive bias.
>
> ---
>
>
>
> RW1. RQ 5. And author refers back to his example in each subsection, crystalizing each step and the relevant mathematical expressions in its context
> >A5. We refered to the example in each subsection and highlighted in the revised paper.
> >
> >1. Each paragraph of each subsection 5.2.
> 2. Below the subsection 5.3, and 5.4.
>
> ---
>
>
> RW1. RQ 6. Figure 2 is supposed to summarize the method and provide a comprehensive view. However, I think that it fails to do so because it is hard to parse, with neither the caption nor the text providing sufficient information to make sense of it.
> >A6.
> >
> >1. We separated the figure into (a) overall architecture, (b) codebook, and (c) Two-step attention to show intuitively.
> 2. We added the example to view the entire model process on caption of Figure 2 in revised paper.
>
> ---

---

> ### Author Response · Authors · 2024-09-19
> **Response to Reviewer 33dK by Authors (2)**
>
> RW1. RQ 7. It is unclear why there is a hyperparameter ϵ weighting $L_{ee}$ and $L_{de}$ in $L_{equiv}$, but no such parameters are introduced within $L_{codebook}$. The authors should clarify this point and justify this design choice.
> >A7.
> We added the hyper parameter ϵ due to the model sensitivity. As shown in Figure 4-a, the Kullback-Leibler divergence convergences to the highest value, when ϵ is large (ϵ=1.0) and it shows less stable results. It implies that the CFASL with larger ϵ struggles with disentanglement learning, as shown in Table 7a. We highlighted this context in Section 6.1, in the paragraph ‘Impact of hyper-parameter tuning’ in the revised paper.
>
> ---
>
> RW1. RQ 8. The contribution in defining $m-FVM_k$ is hard to appreciate without proper background being provided for the vanilla FVM.
> >A8. We add factor-VAE metric on section 2.2, in the paragraph ‘Factor VAE Metric (FVM)’ in the revised version.
>
> ---
>
>
> RW1. RQ 9. Section 5 on related work seems misplaced and should appear as part of the introduction. the authors should include a dedicated paragraph summarizing the specific approach that CFASL will be compared to.
> >A9. We moved the related work section before the Section 4, which define the problem. Also, we summarized the compared methods in Section 3, in the ‘disentanglement learning’ paragraph and the ‘group theory-based approaches for disentangled representation’ paragraph. We highlighted on each paragraph in the revised paper.

---

> > ### Comment · Reviewer_33dK · 2024-10-04
> > **Response to revision**
> >
> > I thank the authors for addressing my comments in the revision. My concerns were largely addressed. One minor suggestion for the final version is to use standard math notation for norms and inner products, i.e., \langle and \rangle instead of < and >, and \|x\| instead of ||x||. This will make the technical content look cleaner.

---

### Review · Reviewer_AYog · 2024-08-22

**Summary Of Contributions:**

The authors propose an unsupervised method for disentanglement in VAE learning, by leveraging symmetries and using tools from group theory. The new method, CFASL, includes several new components that are adequately motivated and whose effects are demonstrated in an ablation study. The authors additionally propose a new evaluation metric. The method is evaluated in several benchmark data sets and shown to outperform previous methods, and the paper includes numerous visualisations illustrating the method.

**Audience:**

Yes

**Claims And Evidence:**

Yes

**Requested Changes:**

I would like to see the writing double-checked for correctness and clarity, but do not see need for any other substantial changes.

Since the authors are anyway short on space, the remark about providing proper background and notation for standard VAE could be done in Supplement; just write in Section 2 that in this paper you assume the reader is well familiar with basic VAE but that you provide a brief summary in Appendix so that the overall objective etc are clear.

**Strengths And Weaknesses:**

Overall, the paper meets the evaluation criteria of TMLR. It is without doubt interesting for a relatively broad audience within the machine community, by the virtue of introducing a new qualitatively different solution for a commonly studied problem. The claims are well supported by empirical evidence and the mathematical reasoning is sound. However, I acknowledge that I have not been able to verify the correctness of the derivations and new method contributions in detail, as someone not particularly familiar with the formalisms used here.

Strengths:
 - The work is well positioned within the related work, including also recent papers, and it includes clear novel elements.
 - The main contributions are clear.
 - The empirical evaluation is thorough, both in terms of comparison methods and the evaluation tasks. Qualitative comparisons are supported with clear illustrations, and already Fig. 1 helps understanding the key qualitative difference well.
- The paper includes several illustrations that help understanding how the method works and the illustrations have been designed carefully to emphasize the main findings.

Weaknesses:
 - The paper is understandable, but the language could be improved in many places. For instance, some sentences are missing a verb. Perhaps consider using proofreading service?
 - The paper is not completely self-contained, for instance not presenting the basic VAE principle and objective at all. The interested readers can be expected to be familiar with VAE, but it would still be better to introduce e.g. the notation used in this work properly. In Section 2.2 you cover previous works by saying they introduce "an additional objective" which is a bit confusing when the main objective has not even been mentioned.
- The mathematical notation could be improved for clarity. At least write "attn" and "softmax" (on page 5) with something like \text{attn}. What is the symbol in Eq. (4)?
- The motivations for the key elements are brief and sometimes missing. For instance, in Section 4.3 you clearly describe what you did but not really why these things were done, or what is the effect of the choices you made (e.g. enforcing commutativeness). Explaining the reasons and their implications in a bit more detail would be useful for the readers.
- The manuscript is somewhat crammed, clearly requiring both compact writing and tricks for figure placement and sizing to fit into the 12 pages. Many of the good illustrations are a bit lost due to being squeezed so small (even the illustrations in Appendix are a bit smallish) the tables are all using someone small font size, and as mentioned above the paper is lacking background and verbal explanations. I think the paper would work better if it was extended to 15-18 pages (without adding new technical content).

---

> ### Author Response · Authors · 2024-09-19
> **Response to Reviewer AYog by Authors (1)**
>
> Dear reviewer AYog,
>
> We appreciate your valuable and fruitful comments to improve this research. The abbreviation ‘RW’ refers to a reviewer, ‘RQ’ refers to a request, and ‘A’ refers to an answer.
>
> ---
>
>
> R2. RQ1. The paper is understandable, but the language could be improved in many places.
> >A1. We modified the typo and grammar.
>
> ---
>
> R2. RQ 2. The paper is not completely self-contained, for instance not presenting the basic VAE principle and objective at all
> >A2. We added β-VAE framework on Section 2.2, in the paragraph ‘Variational Auto-Encoder(VAE)’ in the revised version. We highlighted the paragraph.
>
> ---
>
> R2. RQ 3. it would still be better to introduce e.g. the notation used in this work properly
> >A3. We added the notation table (Table 1 the in revised paper), and we highlighted the table.
>
> ---
>
> R2 RQ 4. In Section 2.2 you cover previous works by saying they introduce "an additional objective" which is a bit confusing when the main objective has not even been mentioned.
> >A4. We added the main objective function as an Equation (1) in Section 2.2, in the ‘variational Auto-encoder(VAE)’ paragraph in the revised paper.
>
> ---
>
> R2 RQ 5. At least write "attn" and "softmax" (on page 5) with something like \text{attn}.
> >A5. We modified these notations on the revised paper.
>
> ---
>
> R2 RQ 6. What is the symbol in Eq. (4)?
> >A6. Equation (4) is a cross-entorpy loss (Equation (7) on the revised paper). We replaced the broken symbol as $L_p = \\sum_{i=1}^{|S|} \\sum_{c \\in C} \\mathbb{1}[T_i=c] \\cdot \\log P(T_i = c| [M; \\Sigma];[\\mathbf{W}_s^i, \mathbf{b}_s^i])$, where $P(T_i = c| [M, \\Sigma];[\\mathbf{W}_s^i, \mathbf{b}_s^i]) = p_s^i$, $p_s^j = [M;\\Sigma] \\mathbf{W}_s^i + \\mathbf{b}_s^i$. We highlighted it in the revised paper (Subsection 5.3, in the ‘second step: section selection’ paragraph).
>
> ---
>
> R2 RQ 7. The motivations for the key elements are brief and sometimes missing. For instance, in Section 4.3 you clearly describe what you did but not really why these things were done, or what is the effect of the choices you made (e.g. enforcing commutativeness). Explaining the reasons and their implications in a bit more detail would be useful for the readers
> >A7.
> >
> >1. We proposed 1) the parallel loss ($L_{pl}$) to give inductive bias that symmetries from the same section $exp(\mathfrak{g}^i_k)$, where $k \in \\{1, 2, \ldots, |SS|\\}$ affect the same factor, 2) the perpendicular loss ($L_{pd}$) that symmetries from the different section $g_c^i$, where $i \in \\{1, 2, \ldots, |S| \\}$ affect different factors, and 3) sparsity loss ($L_{pl}$) that each section changes a single dimension of latent vectors for disentangled representation.
> 2. We added these points on the caption of Figure 2 with details of process and example, also we highlighted them in the revised paper.
> 3. The commutative loss ($L_{pl}$) is the key loss to reduce the computing cost, as we implement the composite symmetry $g_c=\prod_{i=1}^{|S|} \hat{g}_c^i$ . This requires computing the matrix exponential |S| times, which is too costly. To reduce its computing cost, we enforce the commutativity because of the matrix exponential property: $exp⁡(A)  exp⁡(B)=exp⁡(A+B)$ when $AB=BA$. We highlighted this in the revised paper in Section 5.2, in the paragraph ‘Commutativity loss for computational efficiency’.
>
> ---

---

> ### Author Response · Authors · 2024-09-19
> **Response to Reviewer AYog by Authors (2)**
>
> R2 RQ 8. The manuscript is somewhat crammed, clearly requiring both compact writing and tricks for figure placement and sizing to fit into the 12 pages. Many of the good illustrations are a bit lost due to being squeezed so small (even the illustrations in Appendix are a bit smallish) the tables are all using someone small font size.
> >A8-1.
> >1. A pair of images (e.g., $differences$ between two images are in the x- and y-position) is given, and the goal of the model is to represent the differences on the latent vector space, called the composite symmetry $g_c$ for disentangled representations.
> >2. The $codebook$ is designed to represent the composite symmetry $g_c$.
> >3. Each section of the codebook is separated to affect a single factor e.g., the $i^{th}$ section affects the x-position, and the $j^{th}$ section affects the y-position of images.
> >4. Each section consists of Lie algebra to provide diversity of symmetries.
> >5. Each loss optimizes the codebook to guarantee the 3) as follows:
> >    1. Symmetries from the same section affect the same factor e.g., $exp(\mathfrak{g}^i_k)$, where $k \in \\{1, 2, \ldots, |SS|\\}$
> >    2. Each section affects different factors, e.g., $g_c^1, g_c^2, \ldots$ affect x-position, y-position, and, etc.
> >    3. Each section changes a single dimension of latent vectors for disentangled representation.
> >6. $Attn$ ensures diversity in symmetries representation, and $p_s$ predicts the activate dsection, int he case, $i^{th}$ and $j^{th}$ sections for x- and y-position differences.
> >7. The model then represents the composite symmetry $g_c$.
> >8. Lastly, model optimizes the $L_{ee}$ to match $g_c z_1 (=z_2^\prime)$, and $L_{de}$ to match the $x_1$ and $p_\theta(g_c \circ q_\phi(x_2))$ to inject the inductive bias.
>
> >A8-2. We referred the example in each subsection and highlighted in the revised paper.
> >1. Each paragraph of each subsection 5.2.
> >2. Below the subsection 5.3, and 5.4.
>
> >A8-3. We resized the Appendix illustrations.
>
> ---

---

> > ### Comment · Reviewer_AYog · 2024-10-02
> > **Response**
> >
> > Apologies for a late response. I have checked the updated version and read through your comments, and I have no outstanding issues with the paper. The proposed changes address well the remarks I had. The illustrations are still quite small throughout the paper that makes it somewhat tedious to read, but I understand the space is limited.
> >
> > For the very final version I would recommend making the illustrations in the Appendix still larger, since there the space is not so limited. Fig. 9 could well be a bit bigger as the font size is now too small, page 20 layout problem should be fixed, and Fig 16 could be split into 2-3 figures so that e.g. the Car 3D images would not need to be so tiny.

---

### Review · Reviewer_FuzW · 2024-09-08

**Summary Of Contributions:**

I am not very familiar with this theory and lack domain knowledge regarding the experiments.

The paper presents a novel method, Composite Factor-Aligned Symmetry Learning (CFASL), aimed at enhancing disentanglement learning in Variational Autoencoders (VAEs) without requiring prior knowledge of dataset factor information. The authors argue that existing unsupervised methods often depend on known factors, which limits their applicability. CFASL introduces several innovative features, including an explicit learnable symmetry codebook, the concept of composite symmetries, and a group equivariant encoder-decoder architecture. Additionally, the authors propose an extended evaluation metric for assessing multi-factor changes, demonstrating the effectiveness of CFASL through quantitative and qualitative analyses.

**Audience:**

Yes

**Claims And Evidence:**

Yes

**Requested Changes:**

Typo: latnet -> latent

**Strengths And Weaknesses:**

I am not very familiar with this theory and lack domain knowledge regarding the experiments.

My current perspective is, the introduction of CFASL is a significant advancement in the field of disentanglement learning. By removing the dependency on known factors, the method opens new avenues for unsupervised learning.
Innovative Features: The three components of CFASL—inductive bias for latent vector alignment, composite symmetry learning, and group equivariant training—are well-articulated and contribute meaningfully to the overall framework.
Evaluation Metric: The proposed extended evaluation metric for multi-factor changes provides a more comprehensive assessment of disentanglement performance, which is a valuable addition to the existing literature.
Empirical Results: The quantitative results indicate that CFASL significantly outperforms state-of-the-art methods in both single-factor and multi-factor change conditions, supporting the authors' claims about the effectiveness of their approach.
Areas for Improvement.

While the paper introduces several novel concepts, some sections could benefit from greater clarity and detail. For instance, a more thorough explanation of the architecture and training process of the group equivariant encoder-decoder would enhance understanding.

Comparison with Baselines: Although the paper reports improved performance over existing methods, more extensive comparisons with a broader range of baseline models would strengthen the validation of CFASL. Including qualitative comparisons alongside quantitative metrics could provide deeper insights into the model's behavior.
Limitations and Future Work: A discussion on the limitations of the proposed method and potential avenues for future research would be beneficial. Addressing scenarios where CFASL might struggle or fail to perform could provide a more balanced perspective.
Generalization: While the method shows promise, it would be helpful to discuss the generalizability of CFASL across different datasets and contexts. Including experiments on diverse datasets could bolster claims about the method's robustness and versatility.

Conclusion: Overall, the paper presents a compelling and innovative approach to unsupervised disentanglement learning in VAEs through CFASL. The method's novel features and strong empirical results indicate significant contributions to the field. With some improvements in clarity, comparative analysis, and discussions of limitations, this work has the potential to make a notable impact on future research in disentanglement learning.

I recommend accepting this paper for publication, contingent upon addressing the aforementioned areas for improvement.

---

> ### Author Response · Authors · 2024-09-19
> **Response to Reviewer FuzW by Authors (1)**
>
> Dear reviewer FuzW,
>
> We appreciate your valuable and fruitful comments to improve this research. The abbreviation ‘RW’ refers to a reviewer, ‘RQ’ refers to a request, and ‘A’ refers to an answer.
>
> ---
>
> RW3 RQ 1. For instance, a more thorough explanation of the architecture and training process of the group equivariant encoder-decoder would enhance understanding.
> >A1-1. We add the example to view the whole model process on caption of Figure 2 in revised paper. Our model process as follows:
> >1. A pair of images (e.g., $differences$ between two images are in the x- and y-position) is given, and the goal of the model is to represent the differences on the latent vector space, called the composite symmetry $g_c$ for disentangled representations.
> >2. The $codebook$ is designed to represent the composite symmetry $g_c$.
> >3. Each section of the codebook is separated to affect a single factor e.g., the $i^{th}$ section affects the x-position, and the $j^{th}$ section affects the y-position of images.
> >4. Each section consists of Lie algebra to provide diversity of symmetries.
> >5. Each loss optimizes the codebook to guarantee the 3) as follows:
> >    1. Symmetries from the same section affect the same factor e.g., $exp(\mathfrak{g}^i_k)$, where $k \in \\{1, 2, \ldots, |SS|\\}$
> >    2. Each section affects different factors, e.g., $g_c^1, g_c^2, \ldots$ affect x-position, y-position, and, etc.
> >    3. Each section changes a single dimension of latent vectors for disentangled representation.
> >6. $Attn$ ensures diversity in symmetries representation, and $p_s$ predicts the activate dsection, int he case, $i^{th}$ and $j^{th}$ sections for x- and y-position differences.
> >7. The model then represents the composite symmetry $g_c$.
> >8. Lastly, model optimizes the $L_{ee}$ to match $g_c z_1 (=z_2^\prime)$, and $L_{de}$ to match the $x_1$ and $p_\theta(g_c \circ q_\phi(x_2))$ to inject the inductive bias.
>
> >A1-2. 2.	We added the summary of process on the subsection 5.1, 5.2, and 5.3 of the revised paper.
>
> ---
>
> RW3. RQ 2. More extensive comparisons with a broader range of baseline models would strengthen the validation of CFASL.
> >A2. We added the groupified VAE [1] and the results are on the table. Also, we added theses result on the revised paper on section 6.1 and highlighted it.
>
> \\begin{array}{|c|c|c|c|c|c|c|c|c|}
> \\hline
> & G-VAE&OURS & G-VAE&OURS  & G-VAE&OURS  & G-VAE&OURS  \\\\
> \\hline
> & FVM & FVM & MIG & MIG & SAP & SAP & DCI & DCI\\\\
> \\hline
> dSprites & 69.75(\\pm13.66) & \\textbf{82.30}(\\pm5.64) &21.09(\\pm9.20) & \\textbf{33.62}(\\pm8.18) & 5.45(\\pm2.25) & \\textbf{7.28}(\\pm0.63) & 31.08(\\pm10.87) & \\textbf{46.52}(\\pm6.18) \\\\
> 3D Car & 92.34(\\pm2.96) & \\textbf{95.70}(\\pm1.90) & 11.95(\\pm2.16) & \\textbf{18.58}(\\pm1.24) & \\textbf{2.10}(\\pm0.96) & 1.43(\\pm0.18) & 26.91(\\pm6.24) & \\textbf{34.81}(\\pm3.85) \\\\
> smallNORB & 46.64(\\pm1.45) & \\textbf{61.15}(\\pm4.23) & 20.66(\\pm1.22) & \\textbf{22.23}(\\pm0.48) & 10.37(\\pm0.51) & \\textbf{11.12}(\\pm0.48) & \\textbf{27.77}(\\pm0.68) & 24.59(\\pm0.51) \\\\
> \\hline
> \\end{array}
> [1] Tao Yang, Xuanchi Ren, Yuwang Wang, Wenjun Zeng, and Nanning Zheng. Towards building a group-based unsupervised representation disentanglement framework. In International Conference on Learning Representations, 2022.
>
> ---
>
> RW3 RQ3. Including qualitative comparisons alongside quantitative metrics could provide deeper insights into the model's behavior.
> >A3. We selected the best case model, which are the best disentanglement performance for qualitative analysis as shown in Figure 7.
>
> ---
>
> RW3 RQ4. Limitations and Future Work: A discussion on the limitations of the proposed method and potential avenues for future research would be beneficial. Addressing scenarios where CFASL might struggle or fail to perform could provide a more balanced perspective.
> >A4. In the real-world dataset, the variance of the factor is much more complex and has more combinations than the used datasets in this paper (the maximum number of factors is 5). Although our method shows the advance of disentanglement learning in multi-factor change conditions, it remains the generalization in real world datasets or larger dataset. We consider extending the learning of composite symmetries in general conditions. Another drawback is to use of six loss functions, which require more hyper-parameter tuning. As the statistically learned group methods reduce hyper-parameters [2], we consider a more computationally efficient loss function.
> Also, we add it and highlighted on section 8 of the revised paper.
> >
> >[2] Robin Winter, Marco Bertolini, Tuan Le, Frank Noe, and Djork-Arné Clevert. Unsupervised learning of group invariant and equivariant representations. In Alice H. Oh, Alekh Agarwal, Danielle Belgrave, and Kyunghyun Cho (eds.), Advances in Neural Information Processing Systems, 2022.
>
> ---

---

> ### Author Response · Authors · 2024-09-19
> **Response to Reviewer FuzW by Authors (2)**
>
> RW3 RQ 5. Generalization: While the method shows promise, it would be helpful to discuss the generalizability of CFASL across different datasets and contexts. Including experiments on diverse datasets could bolster claims about the method's robustness and versatility.
> >A5. We added the Morpho-MNIST dataset results in Figure 7 to compare the baseline and our method. Also, we added the dataset coverage in Subsection 6.2, in the paragraph ‘factor aligned latent dimension’ and highlighted it in the revised paper.

---

### Decision · Action_Editor_kebR · 2024-11-04

**Recommendation:** Accept as is

**Comment:**

The paper proposes Composite Factor-Aligned Symmetry Learning, which introduces symmetry-based disentanglement into VAE. Although the method is slightly adhoc without theoretical justification, the experiments are broad and clear.

All the reviewers believe this is an interesting piece of work and suggest for acceptance.

**Audience:**

The paper is focusing on disentanglement in VAE, the paper might be interesting to the intersection of communities focusing on group theory and representation learning.

**Claims And Evidence:**

Yes.